



# Inferring the Controlling Factors of Ice Aggregation from Targeted Cloud Seeding Experiments

Huiying Zhang[1], Fabiola Ramelli[1], Christopher Fuchs[1], Nadja Omanovic[1], Anna J. Miller[1], Robert Spirig[1], Zhaolong Wu[2], Yunpei Chu[1], Xia Li[3], Ulrike Lohmann[1], and Jan Henneberger[1]

[1]Institute for Atmospheric and Climate Science, ETH Zurich, Zurich, Switzerland
[2]Leibniz Institute for Tropospheric Research (TROPOS), Leipzig, Germany
[3]Institute for Machine Learning, ETH Zurich, Zurich, Switzerland

**Correspondence:** Huiying Zhang (huiying.zhang@env.ethz.ch) and Jan Henneberger (jan.henneberger@env.ethz.ch)

**Abstract.** Ice aggregation in clouds plays a crucial role in cloud development and precipitation formation. Despite the significance of ice aggregation, direct in situ quantification of aggregation rates in natural clouds has been challenging due to the difficulty of tracking ice crystals. Here, we present in situ measurements of ice aggregation rates in persistent supercooled stratiform clouds. Using novel glaciogenic seeding experiments (CLOUDLAB), ice crystals are nucleated upwind and subse-
quently measured downwind after a known residence time in cloud, allowing us to estimate their age. A deep-learning-based detection algorithm (IceDetectNet) counts the individual monomers of aggregates to derive the initial ice crystal number concentration ($ICNC_{t0}$). We considered several factors that may influence ice aggregation, including $ICNC_{t0}$, temperature, ice crystal size, aspect ratio, and turbulence. Among these, $ICNC_{t0}$ was found to be the dominant factor controlling aggregation rates by three independent approaches: causal inference, a physical equation, and machine learning models. We report,
however, a subquadratic dependence of the aggregation rate on $ICNC_{t0}$ (mean exponent ∼0.92), in contrast to theoretical expectations (quadratic dependence). One possible explanation is that aggregation may also involve smaller ice crystals, but this remains hypothetical. To predict aggregation rates, we evaluated 11 machine learning models and a physically based formulation. CatBoost achieved the best statistical performance, while the physical model proved more robust in sensitivity tests. These findings provide new insights into the microphysical and environmental controls of ice aggregation and establish a robust
methodological foundation for studying aggregation processes in natural clouds.

## 1 Introduction

Ice aggregation is a key microphysical process that influences cloud development and precipitation formation. This process has important implications for weather prediction and climate modeling. During the early stage of ice growth, diffusional processes such as the Wegener–Bergeron–Findeisen mechanism dominate (Korolev, 2007). As clouds mature, collisional pro-
cesses including aggregation and riming become increasingly important (Connolly et al., 2012; Heymsfield, 1986; Sölch and Kärcher, 2011; Hosler et al., 1957). These processes enable ice crystals to grow into larger particles more rapidly than by vapor deposition alone and are fundamental for the formation of snowflakes and graupel, eventually leading to precipitation-sized hydrometeors (Heymsfield, 1986; Sölch and Kärcher, 2011).



Numerous in situ observations have confirmed that ice–ice aggregation occurs in clouds over a wide temperature range,
from just below 0 °C down to −60 °C (e.g., Connolly et al. (2005); Crosier et al. (2011); Field and Heymsfield (2003)).
Aggregated ice crystals often constitute a substantial fraction of total ice. For example, irregular ice crystals—which include
both aggregated and aged ice crystals—have been reported to account for 84 % of total ice crystals in stratiform clouds (Korolev
et al., 2000), 88 % in ground-based snow particle measurements (Zamorsky, 1955), and 94 % in Arctic clouds (Korolev et al.,
1999). More specifically, aggregated ice alone has been observed to comprise 38 % of ice crystals in Arctic mixed-phase clouds
(Zhang et al., 2024), 45 % in thunderstorm clouds (Jaffeux et al., 2022), and 52 % on average across multiple cloud types during
aircraft campaigns (Moss and Johnson, 1994). Despite their apparent ubiquity, quantitative understanding of aggregation rates
remains limited.

Several factors have been hypothesized to influence ice aggregation, such as temperature, ice crystal shape, size, and turbu-
lence. Temperature dependence has long been debated, with laboratory studies conflicting trends: Hosler and Hallgren (1960)
observed a maximum aggregation efficiency near -15 °C, possibly due to the prevalence of dendritic growth forms that interlock
upon collision. In contrast, earlier findings by Hosler et al. (1957) suggested peak aggregation rates around 0 °C to -5 °C, where
quasi-liquid layers on ice surfaces may enhance adhesion (Lamb and Verlinde, 2011). These competing mechanisms—the
habit-based, dendritic ice crystal interlocking mechanism at colder temperatures versus the enhanced surface stickiness near
freezing, which happens at the same time—highlight the complexity of aggregation processes. Turbulence may further enhance
aggregation by introducing small-scale velocity fluctuations that promote collisions (Chellini and Kneifel, 2024; Sheikh et al.,
2022). They also suggest that multiple other pathways may operate, depending on the ambient conditions. Ice crystal size
and number concentration also play important roles. Larger crystals exhibit greater fall-speed differences and larger geometric
cross-sections, while higher ice crystal number concentrations (ICNC) increase collision frequency; both factors enhance the
probability of collision and sticking (Hobbs et al., 1974; Field and Heymsfield, 2003; Field et al., 2006; Connolly et al., 2012;
Karrer et al., 2021). However, disentangling these effects in natural clouds remains challenging due to observational constraints
and the interplay of multiple factors.

Several experimental approaches have been developed to estimate aggregation, yet each carries inherent limitations. Aircraft
measurements, often using a "Lagrangian spiral descent" strategy, infer aggregation efficiency from changes in ice crystal size
distributions as the aircraft descends at approximately the terminal fall speed of the crystals (Field and Heymsfield, 2003;
Field et al., 2006). These measurements rely on 2-D imaging probes and typically assume that decreases in ice crystal number
concentration result from aggregation. However, uncertainties arise from experimental artifacts (e.g., ice crystal breaking up
on the inlets of the probe) and misattribution of number loss to aggregation alone (McFarquhar et al., 2007; Lawson, 2011).
Laboratory studies using ice cloud chambers offer better control over experimental conditions and avoid shattering artifacts,
but chamber dimensions are generally too small to accommodate the timescales required for aggregation to occur naturally
(Shaw et al., 2020). Moreover, both aircraft and laboratory studies typically rely on indirect estimates—based on changes in
ICNC—to infer aggregation efficiency, rather than direct observations (Connolly et al., 2012; Field and Heymsfield, 2003).

These challenges make direct in situ measurements of aggregation rates in natural cloud systems challenging, yet critical for
constraining microphysical parameterizations in models. In this study, we use a novel combination of UAV-based glaciogenic



cloud seeding experiments (Henneberger et al., 2023) within CLOUDLAB project and an advanced deep-learning detection
algorithm (Zhang et al., 2024) to address these gaps. This approach establishes a well-defined initial condition for ice crystal
formation and subsequently captures a detailed snapshot of the ice crystal population after their residence in cloud, thereby
allowing aggregation to be quantified over a precisely constrained time interval. The deep-learning algorithm (IceDetectNet)
robustly identifies individual ice monomers within aggregated ice crystals, providing an unprecedented level of detail for
quantifying aggregation.

The specific research questions that we address are:

1. Which microphysical and meteorological factors control the rate of ice aggregation?

2. How rapidly does aggregation occur following initial ice formation, and to what extent can the aggregation rate be
inferred from the controlling parameters?

To answer these questions, we investigate ice aggregation rates in stratiform clouds using both data-driven and physically
derived approaches. Specifically, we (i) identify the microphysical and meteorological controlling controls on aggregation
(Sect. 3.1); (ii) disentangle the direct and indirect effects of each factor using causal inference (Sect. 3.2); (iii) evaluate predic-
tive models trained on these factors (Sect. 4.2 and Sect. 4.3); and (iv) test the sensitivity of the predictions to temperature and
the initial ice crystal number concentration ($ICNC_{t0}$) (Sect. 4.4).

## 2   The CLOUDLAB campaign and observational data

The CLOUDLAB experiments were conducted in persistent wintertime stratus clouds over the Swiss Plateau, with mea-
surements centered at a site near Eriswil ($47°04'14''$N, $7°52'22''$E; 920 m a.s.l.). These clouds were typically supercooled,
liquid-dominated, and quasi-stationary, with bases below 1,000 m a.g.l. and thicknesses of several hundred meters (Scherrer
and Appenzeller, 2014). We performed glaciogenic seeding upwind of the measurement site using uncrewed aerial vehicles
(UAVs) and sampled the resulting microphysical changes with a suite of in situ and remote sensing instruments. Methods
specific to this study are detailed below, including the seeding operations and instrumentation (Sect. 2.1) and observed ice
properties (Sect. 2.4).

### 2.1   Seeding Operations and Instrumentation

Glaciogenic seeding was performed upwind of the measurement site using a customized uncrewed aerial vehicle (UAV; Meteo-
drone MM-670, Meteomatics AG, Switzerland) equipped with burn-in-place flares (Zeus MK2, Cloud Seeding Technologies,
Germany). Each flare contained approximately 200 g of seeding material, including about 20 g of silver iodide and other ice-
active compounds effective at temperatures below $-5°$C (Chen et al., 2024; Miller et al., 2024). The UAV was operated by
flying multiple crosswind legs (200–400m) while releasing seeding particles for 5–6 minutes at distances of 2–3km upwind of
the measurement site at seeding time $t_0$ (In-cloud seeding and ice nucleation processes in Fig. 1). Further details on the seeding
operations and ice nucleation mechanisms are provided in Miller et al. (2024, 2025) .



The plume of ice crystals generated by seeding arrived at the measurement site after 5–10 minutes depending on the wind speed. Because the plume was typically several hundred meters wide and persisted over the site for several minutes, it could be sampled continuously as it passed, yielding a time series of microphysical observations. The residence time—the time between ice nucleation and observation—was calculated by comparing the times of seeding particles release and first ice detection, and using the observed wind speed across the plume to estimate advection time. To avoid underestimating travel time, the

maximum wind speed among the measured instruments was used in the calculation (See Fuchs et al. (2025) for more details regarding the method). Throughout this growth period, the temperature and wind speed were assumed to remain constant.

The microphysical properties of the plume were measured using a suite of in situ and remote sensing instruments at measurement time $t_1$ (see Fig. 1, Ice detection). Remote sensing measurements included ground-based Ka- and W-band Doppler cloud radars. These radars were used to observe the general cloud structure (e.g., cloud top height) before and after seeding.

They were also used to retrieve the turbulence intensity, which is expressed as the eddy dissipation rate (EDR). EDR was retrieved from the Ka- and W-band radars following the spectral-width method (Appendix E in Wu et al. (2025)). The retrieval assumes homogeneous, isotropic turbulence within the radar volume. Primary EDR estimates were obtained from a mira35 MBR7 Ka-band Doppler radar, with an RPG94 W-band radar used to fill data gaps. Both radars have identical beamwidths (0.5°), with dwell times of 3 s (MBR7) and 5 s (RPG94), and the EDR was averaged over a 30 s temporal window. Horizontal

wind speed, required for the retrieval, was obtained from mira35 MBR5 PPI scans, ECMWF Reanalysis v5 reanalysis, and in situ measurements. The spatial resolution of the retrieved EDR was approximately 30 m per grid cell.

In situ observations were provided by a tethered balloon system (TBS) carrying the HOLographic Imager for Microscopic Observations (HOLIMO; Ramelli et al. 2020). The TBS consisted of a 200 m$^3$ helium-filled kytoon capable of reaching altitudes up to 1 km above ground. Suspended 30 m below the kytoon, the instrument platform carrying HOLIMO, a digital

in-line holography system with resolving sizes of 6 μm for cloud droplets and 25 μm for ice crystals. HOLIMO operated continuously before, during, and after each seeding experiment, recording cloud droplets throughout and ice crystals over the duration of the plume encounter. Holograms were reconstructed using HoloSuite (Fugal and Shaw, 2009) at 1 s resolution, and particles were classified as artifacts, cloud droplets, or ice crystals using a convolutional neural network (Touloupas et al., 2020). Manual verification was performed for all cloud droplets and ice crystals larger than 35 μm to ensure classification

accuracy; no ice crystals smaller than this threshold were observed.

Classified ice crystals were then further analyzed using a fine-tuned version of IceDetectNet (Zhang et al., 2024), named IceDetectNet-CLOUDLAB (see Sect. 2.3 for details). IceDetectNet-CLOUDLAB is a rotated object detection model trained to assign each monomer a shape label comprising its basic habit and microphysical process, and to estimate the number of monomers per ice crystal. The classification scheme includes three basic ice habits—column, plate, and irregular—and one

microphysical process, riming. If an ice crystal contains more than one monomer, it is additionally classified as aggregated. This results in two independent microphysical processes (riming and aggregation) and four possible states for each ice crystal: pristine, rimed, aggregated, and rimed+aggregated. Combining the three habits with these four states yields a total of 12 distinct ice crystal classes. The uncertainty in cloud droplet number concentration is approximately ±5 %, while that for ice crystal



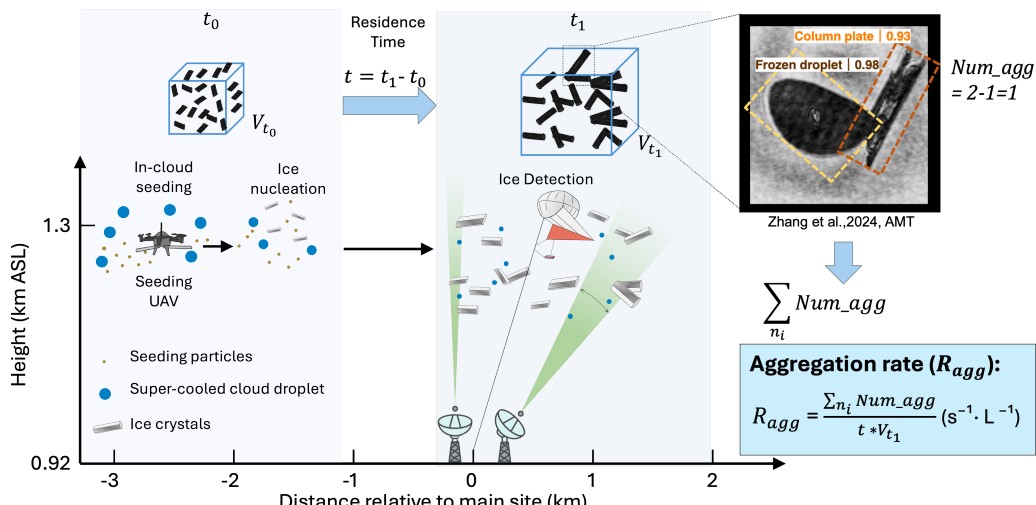

**Figure 1. Scheme for quantifying in situ ice aggregation after cloud seeding.** Seeding particles are released by a UAV, initiating ice crystal formation at time $t_0$. The ice crystals grow and aggregate during the residence time $t_1 - t_0$, and at $t_1$ an image of them is captured by a holographic imager on a tethered balloon system. The ice habit and the number of monomers per aggregate are quantified based on the IceDetectNet (Zhang et al., 2024). For example, the right panel shows an aggregate consisting of two monomers classified as a frozen droplets and a column-plate. The aggregation rate ($R_{agg}$) is defined as the total number of aggregation events, $\sum(n_i - 1)$, divided by the residence time $t$ and the sampled volume $V_{t_1}$ at $t_1$.

number concentration ranges from 5–10 % for crystals larger than 100μm and about 15 % for smaller ones. Uncertainty

quantification for habit/process classification and monomer counts is detailed in Appendix A.

### 2.2 Estimating Aggregation Rates from Experimental Data

Ice aggregation was quantified by estimating the number of monomer-level collisions that formed each observed aggregate. Each detected aggregate containing $n_i$ monomers was assumed to have undergone $(n_i - 1)$ aggregation events and have $n_i$ ice crystals at the initial state $t_0$. The aggregation rate $R_{agg}$ (s$^{-1}$ L$^{-1}$) was thus defined as:

$$R_{agg} = \frac{\sum_{i=1}^{N}(n_i - 1)}{t \cdot V_{t_1}} \tag{1}$$

where $N$ is the total number of detected aggregates, $n_i$ is the monomer count of the $i$-th aggregate, $t$ is the residence time estimated based on the method in (Fuchs et al., 2025), and $V_{t_1}$ is the sampled cloud volume at time $t_1$. $R_{agg}$ represents the mean aggregation rate integrated over the residence time $t$. The approach relies on the following assumptions:



1. **Steady background conditions.** Environmental and microphysical properties of the background cloud (e.g., temperature, wind speed) are assumed to remain constant throughout each experiment.

2. **Instantaneous nucleation.** Ice nucleation is assumed to occur immediately upon release of seeding particles, providing a well-defined initial time to ice formation. Our observations have revealed that ice formation is likely initiated very quickly after the release of the highly hygroscopic and ice-active seeding material (Miller et al., 2025).

3. **Conservation of monomer number.** The total number of ice crystal monomers is assumed to be conserved between the seeding and observation locations. This implies that (i) sedimentation losses within the plume are balanced by ice crystals falling from above, which is supported by radar observations and (ii) no secondary ice production (SIP) occurs. The latter is supported by observations: no large droplets or graupel were detected, and no evidence of frozen droplet breakup was observed, consistent with conditions unfavorable for SIP.

4. **Complete detection of aggregate monomers.** All monomers within an aggregate are assumed to be accurately identified by IceDetectNet. However, some early aggregates may become unresolved after diffusional growth and be classified as irregular, but this fraction is very small (see Fig. 2h). Other potential detection uncertainties and classification errors are discussed in Appendix A and are considered negligible.

### 2.3 Training IceDetectNet-CLOUDLAB for Aggregated Ice Monomer Identification

To retrieve the number of monomers in each aggregate, we fine-tuned the original IceDetectNet model (Zhang et al., 2024) to create IceDetectNet-CLOUDLAB, specifically adapted for holographic images collected during CLOUDLAB seeding experiments. The original model was trained to classify ten ice crystal habits; for this application, the output layer was reconfigured to distinguish four categories: column, plate, column-rimed, and plate-rimed. The final classification layer was randomly initialized, while all other parameters were taken from the IceDetectNet. A total of 2,380 manually labeled images were used, randomly selected to ensure an approximate balance across all seeding experiments. Of these, 2,134 images were used for training and 246 for testing. The model architecture followed IceDetectNet-CLOUDLAB (Zhang et al., 2024), based on S2ANet with a ResNet-50 backbone. To improve training stability on the smaller CLOUDLAB dataset, which contains fewer ice crystals and fewer classes than previous applications, the initial learning rate was reduced to 0.0001. The number of training epochs was extended to 200, with learning rate decay scheduled earlier at epochs 32 and 48 to encourage earlier convergence and to prevent overfitting. A linear warmup phase of 1,000 iterations was applied to avoid early gradient instability. During inference, up to six predictions per image were retained after non-maximum suppression (IoU threshold = 0.5).

### 2.4 Statistical Characterization of Ice Crystal Properties

We analyzed 21 seeding experiments (Table F1) conducted at temperatures between $-4.7\,^{\circ}\mathrm{C}$ and $-7.8\,^{\circ}\mathrm{C}$. The ice crystal habit distribution differed systematically with temperature: warmer experiments ($T > -7\,^{\circ}\mathrm{C}$) contained exclusively columnar crystals, whereas the colder experiments ($T \leq -7\,^{\circ}\mathrm{C}$) contained both plates and short columns (Fig.E1). Plates and columns





have been shown to exhibit different diffusional growth rates along their major axes (Fuchs et al., 2025), which can influence subsequent aggregation. Based on this habit difference, the experiments were classified into "warmer" ($T > -7°C$, $n = 14$) and "colder" ($T \leq -7°C$, $n = 7$) regimes (Fig. 2a). Residence times were generally shorter in the colder experiments, ranging from 294–532 s, compared to 371–629 s in the warmer experiments (Fig. 2b).

ICNC$t0$, the initial ice crystal number concentration, was estimated by assuming that each aggregated monomer corresponds
to one ice crystal present at $t_0$. This allows estimation of the ICNC$_{t0}$ at the initial state $t_0$. ICNC$_{t0}$ was higher in the warmer regime than in the colder, with mean ± standard deviation of approximately 815 L$^{-1}$ ± 394 L$^{-1}$ and 566 L$^{-1}$ ± 291 L$^{-1}$, respectively (Fig. 2c). These levels lie between typical ICNC$_{t0}$ values reported for deep convection (Heymsfield and Willis, 2014) ($\sim$100 L$^{-1}$) and those associated with secondary ice production in convective systems (Korolev et al., 2020) ($\sim$1000 L$^{-1}$). Aggregates were more frequent in the warmer regime, with fractions reaching up to 43% of total observed ice crystals, while
remaining around 15% in the colder regime (Fig. 2d). Ice crystals in the warmer regime reach mean major size (defined as the length of the crystal's major axis) per experiment of up to 312 $\mu$m (minimum 101 $\mu$m), compared to a range of 66–129 $\mu$m in the colder regime; the warmer regime also exhibited a broader, long-tailed size distribution (Fig. 2e). Aspect ratio (defined as the ratio of the major to minor axis lengths) distributions differed as well: the colder regime was narrowly peaked around 1.4–1.6, while the warmer regime showed a broader and flatter distribution around 1.73-4.13 (Fig. 2f). The riming ratio (RR)
is defined as the fraction of rimed crystals. Rimed crystals were abundant in both regimes, with fractions exceeding 38% in all experiments and reaching up to 94% in one single experiment, indicating that riming was nearly ubiquitous (Fig. 2g). Finally, irregular ice crystals, which have no identifiable habit, were uncommon overall (2–5%; Fig. 2h). Some early aggregates may have grown sufficiently by diffusion to obscure their monomer structure, thus being classified as irregular. These crystals were slightly more frequent in the colder regime, suggesting that any resulting underestimation of aggregation is likely negligible.

## 3 Microphysical and Environmental Controls on Ice Aggregation

We examined the microphysical and environmental controls on ice aggregation through a structured sequence of analyses. We first examined how microphysical and environmental factors influence the observed aggregation rates (Sect. 3.1). We then applied a causal graph to disentangle the direct and indirect effects of these factors on aggregation (Sect. 3.2).

### 3.1 Correlations Between Aggregation Rate and Microphysical and Environmental Factors

To better understand the mechanisms underlying aggregation, we first examine the Pearson correlations between the aggregation rate and four factors: ICNC$_{t0}$, residence time, temperature, and EDR. No clear correlation was observed between aggregation rate and EDR, which characterizes the turbulence intensity within the cloud (see Appendix B for the EDR correlation analysis). This suggests that, under the present experimental conditions, turbulence at the resolved scales (30 m × 30 m) did not significantly influence aggregation. The correlations with ICNC$_{t0}$, residence time, and temperature are all presented in the
following.



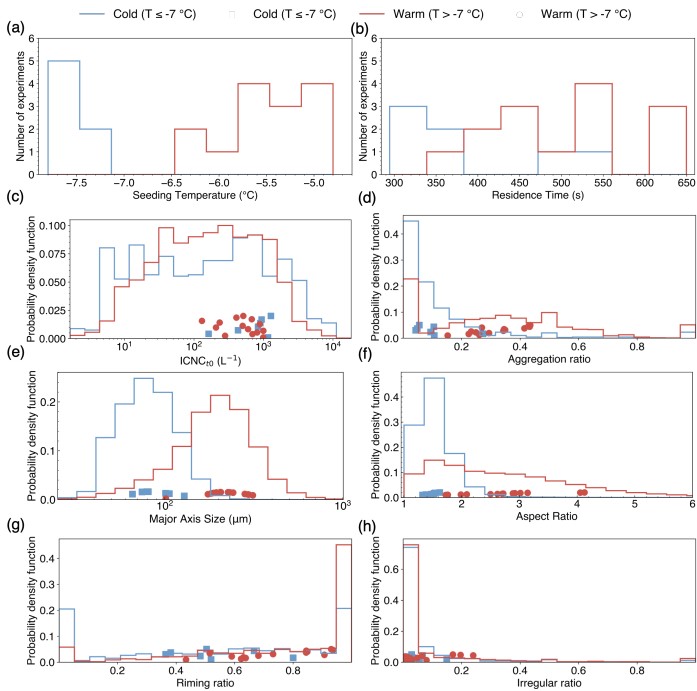

**Figure 2. Distributions of ice crystal properties during seeded periods, grouped by temperature:** colder ($T \leq -7^\circ$C, blue) and warmer ($T > -7^\circ$C, red). (a) Temperature; (b) Residence time; (c) $ICNC_{t0}$; (d) Aggregation ratio; (e) Major size of ice crystals; (f) Aspect ratio; (g) Riming ratio; (h) Irregular ratio. For (a) and (b), the solid lines show the per-experiment distributions. For (c) through (h), the solid lines represent one-second resolution distributions within each regime, and the scatter points indicate the means of the experiments (The scatter positions along the y-axis are offset solely to reduce overlap. Only the values along the x-axis carry meaning).

### 3.1.1 Strong positive correlation between $ICNC_{t0}$ and aggregation rate

$ICNC_{t0}$ exhibited a strong positive correlation with the aggregation rate in all experiments (Fig. D1). This is consistent with theoretical expectations, as higher crystal concentrations increase the probability of collisions leading to aggregation (Hobbs et al., 1974).

To illustrate this relationship more clearly, we present two experiments for reference: a warmer case (SM054, T = $-5\,^\circ$C) and a colder case (SM069, T = $-7.6\,^\circ$C) (Fig. 3a,b). Under the condition that the ice crystals are homogeneously distributed, the aggregation rate (Seifert and Beheng (2006) Eq.62), which is the number of aggregation events per unit time per unit





volume of cloud , can be expressed as:

$$R_{\mathrm{agg}} = \frac{1}{2} K(D_1, D_2, T) \, N_1 \, N_2,$$

where $N_1$ and $N_2$ are the number concentrations of two ice crystal populations of sizes $D_1$ and $D_2$, $T$ is temperature, and $K(D_1, D_2, T)$ is the collision kernel, which depends on ice crystal size, shape, fall speed, and ambient conditions (Connolly et al., 2012). In the simplest case where the population is monodisperse and $N_1 = N_2 = \mathrm{ICNC}_{t0}$, the rate becomes proportional to $\mathrm{ICNC}_{t0}^2$, shown as: $R_{\mathrm{agg}} = \frac{1}{2} K(D_1, D_2, T)(\mathrm{ICNC}_{t0})^2$. This quadratic dependence arises because the collision frequency between ice crystals scales with the product of their number concentrations.

We first evaluated the data against this fixed quadratic relationship ($R_{\mathrm{agg}} = a \cdot \mathrm{ICNC}_{t0}^2$), where the prefactor $a$ effectively represents $\frac{1}{2} K(D_1, D_2, T)$. To account for differences between experiments, $a$ was fitted independently in each case. While this model gets correlation coefficients of $r = 0.93$ and $r = 0.81$ for the warmer and colder cases, respectively, it systematically underestimated aggregation at low $\mathrm{ICNC}_{t0}$ (Fig. 3a,b, dashed grey lines).

A better fit was achieved using a free power-law relationship ($R_{\mathrm{agg}} = a \cdot \mathrm{ICNC}_{t0}^n$), where both $a$ and $n$ were estimated sep-
arately for each experiment. This empirical model closely matched the observations across the full range of concentrations (Fig. 3a,b, solid lines). In the warmer-case experiment (SM054), the aggregation rate scaled with an exponent of $1.09$, improving the correlation to $r = 0.98$ (Fig. 3a). In the colder-case experiment (SM069), the best-fit exponent was $0.68$, with a corresponding correlation of $r = 0.83$ (Fig. 3b). This pattern was consistent across all 21 experiments, where the free-fit power-law systematically performed better than the fixed-quadratic form (see Fig. D1). The average exponent across all ex-
periments was $0.92 \pm 0.11$, smaller than the quadratic dependence expected from the collision kernel formulation (Melzak, 1957; Connolly et al., 2012; Seifert and Beheng, 2006). Bulk microphysics schemes such as Lin et al. (1983) and Morrison and Milbrandt (2015) have even adopted a linear dependence on ICNC, which is closer to our observations, often with an additional size threshold so that aggregation only occurs once ice crystals exceed a certain size. However, our observations hint that aggregation may also occur among smaller ice crystals. This could result in effective dependence with an exponent smaller
than one.

### 3.1.2   No significant correlation between aggregation rate and residence time

Residence time in our experiments ranged from 294 s to 650 s (Table F1, Fig. 4) and showed no significant correlation with aggregation rate ($r = -0.09$, $p = 0.686$). This parameter is specific to our experimental design, as the ice crystals were artificially generated by cloud seeding, enabling an explicit measurement of the total growth period. In natural-cloud studies,
residence time is rarely quantified, and direct comparisons are therefore not straightforward. Theoretically, a longer residence time would be expected to yield a smaller time-averaged aggregation rate, because ICNC decreases over time. We find only a very weak tendency for $R_{\mathrm{agg}}$ to decrease with residence time (Fig. 4, $R^2 = 0.009$), suggesting that our results remain broadly comparable to previous observational and laboratory studies despite this experimental uniqueness (Connolly et al., 2012; Field and Heymsfield, 2003; Field et al., 2006). This weak dependence between aggregation rate and residence time may reflect the
fact that ice crystals must first grow to sufficiently large sizes before collisional aggregation becomes efficient. During the early





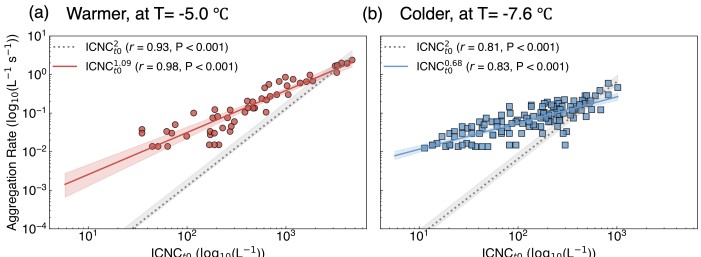

**Figure 3. Correlation between aggregation rate and ICNC$_{t0}$ in two temperature regimes.** (**a**) warmer case (SM054, T $= -5.0\,^{\circ}$C); (**b**) colder case (SM069, T $= -7.6\,^{\circ}$C). One-second data points are shown as red circles (warmer) and blue squares (colder). Solid lines with shaded areas show free power-law fits $R_{\text{agg}} = a \cdot \text{ICNC}_{t0}^{n}$ with 95% confidence intervals (red for warmer, blue for colder); dashed lines with grey shading show quadratic fits ($n = 2$). The prefactor $a$ was fitted independently for each experiment and reflects the effective collision kernel.

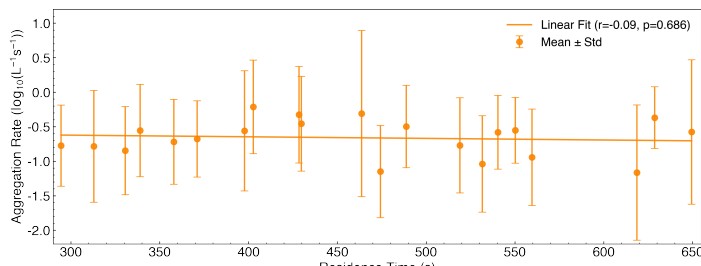

**Figure 4. Correlation between aggregation rate and residence time.** Orange markers with error bars show the mean $\pm$ standard deviation for each experiment, and the solid line represents the linear fit ($r = 0.09$, $p = 0.686$) across all 21 experiments.

growth phase, ICNC decreases rapidly due to plume dilution (as seeded particles are initially concentrated and then disperse; see Fig. 2 in Ramelli et al. (2024)), but this shows little impact on the aggregation rate, making it appear largely insensitive to residence time.

### 3.1.3 Temperature dependence of aggregation rate associated with ice crystal size and aspect ratio

Temperature exhibited a weak positive correlation with aggregation rate across all experiments ($r = 0.43$, $p = 0.053$; Fig. 5).
Although the statistical evidence was weak ($p = 0.053$), the observed trend is consistent with theoretical expectations and previous findings that warmer conditions enhance aggregation (Hosler et al., 1957). Consistent with known temperature-dependent

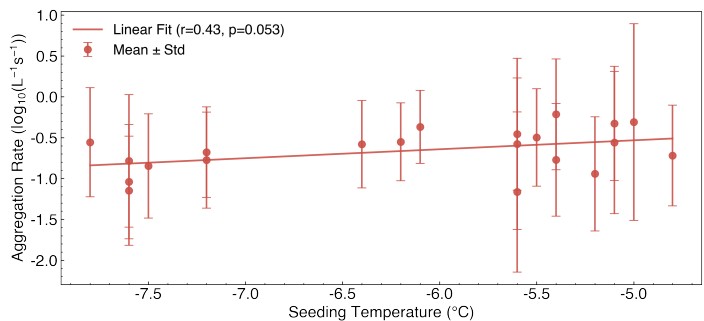

**Figure 5. Correlation between aggregation rate and temperature** Same as Figure 3, but for the Temperature.

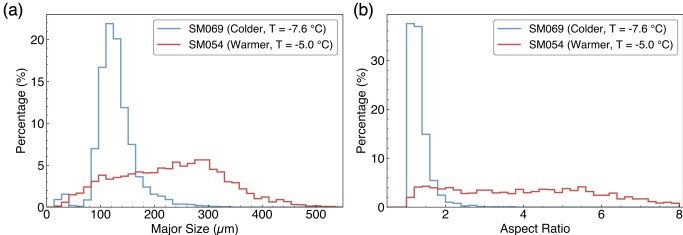

**Figure 6. Distributions of ice crystal major axis length and aspect ratio in warmer and colder cases.** (**a**) Major axis length and (**b**) aspect ratio for SM054 (warmer, red) and SM069 (colder, blue).

habits, the warmer group (above $-7°C$) consisted exclusively of columnar ice crystals (Fig. E1a), whereas the colder group (below $-7°C$) included both columnar and plate-like crystals (Fig. E1b).

To investigate the mechanisms underlying temperature relationship, we compared the distributions of ice crystal size and geometry (the same cases as in Fig. 3), to better understand why the warmer case (SM054) has a higher aggregation rate than the colder case (SM069). Warmer case (SM054) exhibited broader distributions of both major axis length and aspect ratio compared to SM069 (colder case), as shown in Fig. 6. This pattern was not unique to SM054; broader distributions of major size (Fig. 2 (e)) and aspect ratio (Fig. 2 (f)) were consistently observed in warmer experiments. Such variability in shape and

size increases the range of fall velocities among crystals (Heymsfield, 1972; Mitchell, 1996), which promotes more frequent collisions and subsequent aggregation at warmer temperatures.



### 3.2 Causal Pathways Among Aggregation Drivers

Previous analyses showed that $ICNC_{t0}$, temperature, major axis length, and aspect ratio each correlate with aggregation rate to varying degrees. However, these factors are not independent, and their relative contributions and interactions remain unclear.

To disentangle direct and indirect effects among these variables, we constructed a causal graph in the form of a directed acyclic graph (DAG; Fig. 7), incorporating these 4 variables as well as riming ratio (RR), which is the percentage of rimed ice crystals to all the observed ice crystals. Because RR can influence ice crystal geometry — particularly major size and aspect ratio — it may therefore affect aggregation indirectly.

DAGs represent hypothesized cause–effect relationships, with nodes denoting variables and arrows indicating the direction

of influence (Pearl, 2009; Peters et al., 2016). Unlike correlation-based analyses, DAGs explicitly separate direct effects from mediated pathways, enabling quantitative decomposition of total effects. For example, temperature may influence aggregation rate both directly and indirectly by altering ice crystal size and aspect ratio.

The graph structure was specified by combining prior physical knowledge with statistical dependencies inferred from the data. Path coefficients were estimated using structural equation modeling (SEM), which fits a system of regression equations to

quantify the strength of each link. All variables were standardized prior to fitting, so that each coefficient represents the change in aggregation rate (in standard deviations) associated with a one-standard-deviation increase in the predictor, reflecting their relative importance.

The fitted DAG (Fig. 7) shows that $ICNC_{t0}$ (+0.81), temperature (+0.24), and major size (+0.08) have positive direct effects on aggregation rate, with $ICNC_{t0}$ clearly dominating. Aspect ratio exhibits a weak negative direct effect (−0.16), likely

because stronger aggregation tends to increase the number of monomers per particle, producing more compact and rounded aggregates and thus reducing the aspect ratio. Beyond its direct effect, temperature also strongly influences aspect ratio (+0.61) and major size (+0.40), which in turn affect aggregation. Since aspect ratio and major size have opposing effects (negative and positive, respectively), these indirect contributions partially cancel, and temperature maintains an overall positive influence.

RR has a negligible direct effect on aggregation (-0.01). RR indirectly increases the major axis length (+0.31), promoting

aggregation. However, it also decreases $ICNC_{t0}$ (−0.23), suppressing aggregation. These opposing effects cancel each other out, resulting in a minimal net effect of RR on aggregation. Specifically, RR tends to produce larger, more heavily rimed crystals but in lower numbers, as enhanced sedimentation reduces concentrations within the plume. As a result, the net effect of RR remains minimal, suggesting that riming and aggregation are largely independent processes in these experiments.

### 4   Aggregation Rate Predictive Models

Building on the preceding analysis, we next evaluated whether aggregation rates can be quantitatively predicted from the identified microphysical and meteorological factors. We first describe the training procedures and evaluation metrics (Sect. 4.1). Then, we implemented and compared eleven machine learning models (Sect. 4.2) and one physically based equation (Sect. 4.3), using temperature, major axis length, aspect ratio, and $ICNC_{t0}$ as input features and the aggregation rate as the target variable. RR was excluded from the predictive modeling as its direct effect on aggregation was found to be negligible in our causal





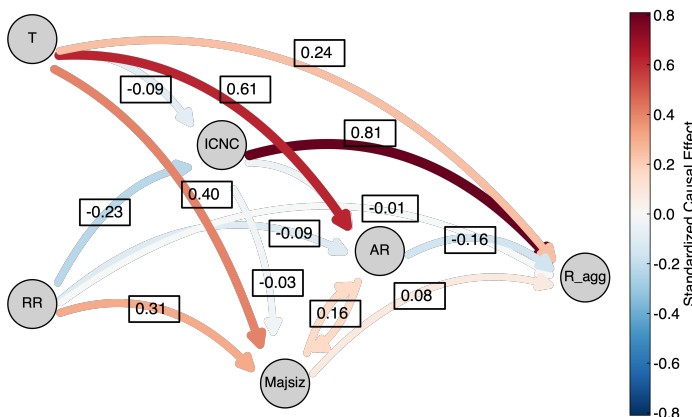

**Figure 7. Causal graph of factors influencing aggregation rate.** The graph depicts the inferred direct and indirect effects on aggregation rate (R_agg) among five variables: $ICNC_{t0}$, temperature (T), major axis length (Majsiz), aspect ratio (AR), and riming ratio (RR). Nodes represent variables, and directed arrows indicate causal relationships. Arrow thickness, color, and the numerical labels on the arrows denote the magnitude and direction of the standardized effects (change in aggregation rate in standard deviations per standard deviation change in the predictor).

inference analysis (Sect.3.2). Finally, to evaluate the robustness of these models under varying atmospheric and microphysical conditions, we performed a sensitivity analysis with respect to temperature and $ICNC_{t0}$ (Sect. 4.4).

### 4.1   Predictive Model Training and Evaluation

We trained the models using temperature, major axis length, aspect ratio, and $ICNC_{t0}$ as predictors of the aggregation rate. All variables except temperature were available at 1 s resolution; temperature was constant for each experiment. We fitted

the analytical model (Eq. 2) by minimizing the mean absolute error (L1 norm) between predicted and observed aggregation rates, with $L_2$ regularization applied to all parameters (penalty coefficient = $10^{-4}$). All coefficients were initialized to one and optimized using gradient descent with a learning rate of $10^{-2}$ over 1500 iterations. All machine learning models were trained with default hyperparameters, except for a fixed random seed (42) to ensure reproducibility. No early stopping, feature engineering, or categorical coding was applied, as all inputs were continuous. Model performance was assessed using five-

fold cross-validation with group-based splits defined by seeding experiment identifiers. This ensured that data from a given experiment were restricted to either training or test sets. For each model and fold, we computed the root mean square error (RMSE), which quantifies the typical magnitude of prediction error, and the coefficient of determination ($R^2$), which reflects the proportion of variance explained. We reported the mean and standard deviation of both metrics across the five folds. After evaluation, each model was retrained on the full dataset and archived for reproducibility. Model interpretability was assessed



using SHapley Additive exPlanations (SHAP). SHAP values quantify how much each feature shifts a prediction away from the dataset's average prediction, with positive values indicating an increase and negative values a decrease. Conceptually, each feature is treated as a "player" in a cooperative game, and its SHAP value represents the average change in the prediction when the feature is added to all possible subsets of other features. This provides a consistent measure of each feature's contribution to the prediction, accounting for interactions with other features. Global feature importance was obtained by averaging the

absolute SHAP values across all samples, and the results were visualized using summary plots.

### 4.2 Machine Learning Models for Aggregation Rate Prediction

#### 4.2.1 Overview of Machine Learning Algorithms

We implemented eleven supervised regression algorithms to model the relationship between environmental and microphysical predictors and aggregation rate. These included linear methods (Linear, Ridge, and Bayesian Ridge Regression), instance-based

learning (K-Nearest Neighbors), decision tree–based models (Decision Tree, Extremely Randomized Trees), and ensemble boosting techniques (Adaptive Boosting, Gradient Boosting, Light Gradient Boosting Machine, Extreme Gradient Boosting, and gradient boosting with categorical features support (CatBoost)). Linear models assume independent, additive effects and are limited in representing nonlinear interactions. Tree-based models capture nonlinear and interaction effects through hierarchical partitioning of the feature space, while boosting methods iteratively refine predictions by combining multiple weak

learners. Detailed descriptions of all models are provided in Appendix C. Among these, CatBoost is particularly well-suited for the current problem because of its ability to handle nonlinear interactions and its robustness to overfitting. CatBoost is a gradient boosting method that uses symmetric (oblivious) decision trees, where all nodes at the same depth split on the same feature and threshold. This symmetry simplifies optimization and improves generalization. It also introduces "ordered boosting," which builds trees in a way that avoids using future information during training, thereby reducing overfitting. Additionally,

CatBoost efficiently handles categorical features and typically requires minimal hyperparameter tuning.

#### 4.2.2 Predictive Performance and Feature Contributions

Among all machine learning models, CatBoost and Gradient Boosting achieved the highest performance, with the lowest RMSE $(1.53 \times 10^{-1}$ s$^{-1}$ L$^{-1}$) and high $R^2$ (0.87) for CatBoost, and a slightly higher RMSE but marginally better $R^2$ (0.88) for Gradient Boosting. Their comparable skill likely reflects their shared ensemble structure and residual learning strategy. Extra

Trees and Light Gradient Boosting Machine followed closely in performance (Table 1). For subsequent analyses, CatBoost was selected as the representative data-driven model owing to its competitive accuracy and robust generalization. The physically derived equation also performed comparably to the tree-based models, achieving an RMSE of $2.10 \times 10^{-1}$ s$^{-1}$ L$^{-1}$ and $R^2$ of 0.78, suggesting that it captures the dominant aggregation dependencies despite its reduced complexity and lack of data-driven tuning. In contrast, linear models (Linear Regression, Ridge, Bayesian Ridge) performed poorly, with $R^2$ values around

0.56 and RMSE around $2.73 \times 10^{-1}$ s$^{-1}$ L$^{-1}$, highlighting the importance of nonlinear interactions and feature dependencies that linear approaches cannot represent. In each experiment (Fig. 8), the agreement between observed aggregation rates and



**Table 1. Model performance comparison.** Mean RMSE ($10^{-1}$ s$^{-1}$ L$^{-1}$) and R$^2$ values along with their standard deviations, sorted by RMSE in ascending order.

| Model | RMSE ($\times 10^{-1}$) | R$^2$ |
|---|---|---|
| CatBoost | $1.53 \pm 1.08$ | $0.87 \pm 0.08$ |
| Gradient Boosting | $1.55 \pm 0.91$ | $0.88 \pm 0.05$ |
| Extremely Randomized Trees | $1.62 \pm 0.94$ | $0.87 \pm 0.05$ |
| Light Gradient Boosting Machine | $1.63 \pm 1.00$ | $0.86 \pm 0.07$ |
| Extreme Gradient Boosting | $1.67 \pm 1.04$ | $0.86 \pm 0.07$ |
| Decision Tree | $1.99 \pm 0.92$ | $0.80 \pm 0.04$ |
| Adaptive Boosting | $2.05 \pm 0.75$ | $0.75 \pm 0.11$ |
| K-Nearest Neighbors | $2.09 \pm 1.05$ | $0.78 \pm 0.06$ |
| Physical Equation | $2.10 \pm 1.34$ | $0.78 \pm 0.11$ |
| Bayesian Ridge Regression | $2.73 \pm 0.90$ | $0.56 \pm 0.16$ |
| Linear Regression | $2.73 \pm 0.90$ | $0.56 \pm 0.16$ |
| Ridge Regression | $2.73 \pm 0.90$ | $0.56 \pm 0.16$ |

CatBoost predictions is evident, with residuals having means near zero and relatively small standard deviations. These results demonstrate that the model effectively captures the relevant physical mechanisms.

To interpret the predictions, we computed SHAP, which quantifies the relative contribution of each feature factor. ICNC$_{t0}$
was the dominant predictor, explaining 69.9% of the variance (Fig. 9), consistent with the earlier correlation analysis (Fig. 3, Fig. D1) and the causal graph (Fig. 7), where ICNC$_{t0}$ also showed the strongest standardized effect. However, a simple linear regression model using ICNC$_{t0}$ alone performed poorly ($R^2 = 0.56 \pm 0.16$, RMSE $= 2.73 \times 10^{-4} \pm 0.90 \times 10^{-1}$ s$^{-1}$ L$^{-1}$; Table 1), underscoring the importance of nonlinear interactions among multiple variables.

Major axis length (14.9%) also contributed meaningfully to the predictions. Larger major sizes were associated with higher
aggregation rates, consistent with the expectation that larger crystals, with greater cross-sectional area and fall-speed variability, enhance collision likelihood (Heymsfield and Miloshevich, 2003) — in agreement with the positive direct effect identified in the causal graph ($+0.08$, Fig. 7). Aspect ratio (8.3%) had a weaker and more complex effect: SHAP values clustered near zero, with a slight tendency for lower aspect ratios to favor aggregation and higher aspect ratios to suppress it, consistent with its negative effect in the causal graph ($-0.16$).

Temperature accounted for 6.9% of the model output. Higher temperatures generally promoted aggregation, consistent with laboratory findings (Hosler et al., 1957) and the causal graph ($+0.24$, Fig. 7). Also, unlike the other variables, temperature was measured as a single value per experiment rather than at a 1-second resolution, which may have limited its explanatory power in the model. Nevertheless, its limited contribution suggests that its influence is largely indirect and mediated by changes in




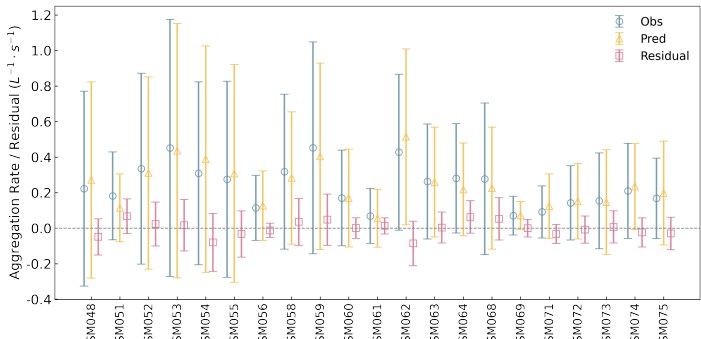

**Figure 8. Observed, predicted, and residual aggregation rates per experiment.** For each experiment, observed aggregation rates (dark blue circles), CatBoost-predicted aggregation rates (yellow triangles), and residuals (pink squares, observed minus predicted) are shown side by side, with observed values on the left, predicted in the middle and residuals on the right. Markers indicate mean values, and error bars represent the standard deviation within each experiment. Rates are expressed in $L^{-1}s^{-1}$.

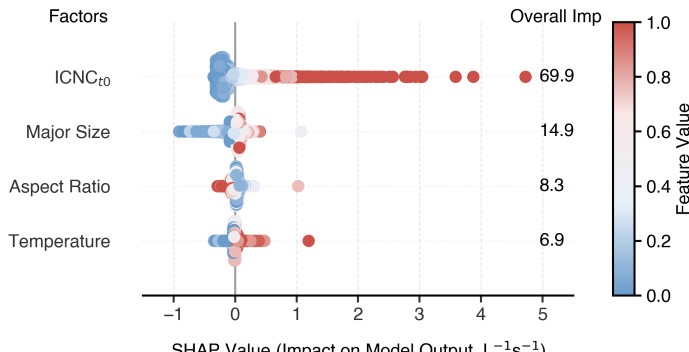

**Figure 9. SHAP analysis of feature contributions to CatBoost model predictions.** Each dot represents one prediction at 1-second resolution from the test data. SHAP values (horizontal axis) indicate the impact of each input feature on the predicted aggregation rate. Features (i.e. $ICNC_{t0}$, Major size, Aspect ratio, and Temperature) are ranked by their overall importance (right), with percentages representing the mean absolute SHAP value normalized across all features. Dot colors reflect the corresponding feature values, with red denoting higher and blue denoting lower values of the feature.

crystal habit and size, as reflected by aspect ratio and major axis length. Therefore, temperature likely acts as a secondary driver

embedded within other structural parameters.





### 4.3 Physical Model

#### 4.3.1 Model Formulation

We constructed an empirical formulation for the ice aggregation rate based on established microphysical principles. The basic structure follows the collision kernel framework (Melzak, 1957), in which the aggregation rate is proportional to the product of the number concentrations of the interacting particles. Under the simplest assumption of equal-sized populations, this scaling becomes quadratic in $\text{ICNC}_{t0}$. However, bulk microphysics schemes such as Lin et al. (1983); Morrison and Milbrandt (2015) have used a linear dependence. To allow flexibility between these theoretical and parameterization-based scalings, the exponent on $\text{ICNC}_{t0}$ was treated as a free parameter. The temperature dependence is represented as an exponential term, consistent with parameterizations of collision efficiency in two-moment schemes (e.g. Seifert and Beheng (2006)), where temperature modulates the quasi-liquid layer and hence the sticking probability upon collision. The dependence on major axis length is formulated as a power law to account for its role in determining both geometric cross-section and differential fall speed, which are classical components of aggregation kernels. Thus, the proposed equation is:

$$R_{\text{agg}} = \alpha \cdot \text{ICNC}_{t0}{}^{\beta_0} \cdot exp^{\beta_1 T} \cdot \text{MajSiz}^{\beta_2} \tag{2}$$

where: $R_{\text{agg}}$ is the observed aggregation rate ($\text{s}^{-1}\,\text{L}^{-1}$), $\alpha$ is a dimensioned scaling constant, $\beta_0$, $\beta_1$, and $\beta_2$ are empirically fitted exponents, $\text{ICNC}_{t0}$ is the initial ice crystal number concentration ($\text{L}^{-1}$), $T$ is the ambient temperature (°C), MajSiz is the average major axis length of ice crystals (m). The term $\text{ICNC}_{t0}{}^{\beta_0}$ accounts for the increased probability of collisions as a function of ice crystal concentration. Collection theory suggests a near-quadratic dependence under the assumptions of random motion and homogeneous mixing (Connolly et al., 2012). We retain a flexible exponent $\beta_0$ to account for inhomogeneities and dispersion effects in real clouds. The temperature-dependent exponential term, $e^{\beta_1 T}$, reflects the nonlinear role of temperature in aggregation-relevant processes. Temperature modulates diffusional growth, growing dimension and the thickness of the quasi-liquid layer, which shapes ice size and habit. Many of these processes exhibit exponential or Arrhenius-like behavior with temperature. The exponential formulation is consistent with aggregation parameterizations in operational two-moment microphysics schemes (e.g. Seifert and Beheng (2006); Lin et al. (1983). The power-law dependence on ice crystal size, $\text{MajSiz}^{\beta_2}$, represents the combined influence of fall speed variability and geometric collision cross section. Larger crystals sediment faster and offer larger interaction surfaces, increasing the likelihood of collision and adhesion. This formulation follows classical approaches to the collection kernel under differential sedimentation. Aspect ratio was excluded from the final model, as it is not explicitly represented in two-moment bulk schemes (Lohmann and Roeckner, 1996; Seifert and Beheng, 2006), and its fitted coefficient in our analysis was negligible compared to the other factors.

The final fitted form is:

$$R_{\text{agg}} = 0.0058 \cdot \text{ICNC}_{t0}{}^{0.73} \cdot \exp(0.18 \cdot T) \cdot \text{MajSiz}^{0.16}, \tag{3}$$





When optimized on the same dataset, this physical model achieved an RMSE of $2.10 \times 10^{-4}$ s$^{-1}$ L$^{-1}$ and an $R^2$ of 0.78 (Table 1). Although lower than the CatBoost model ($R^2 = 0.87$), its performance remains strong, suggesting that the dominant dependencies of aggregation are well captured.

The fitted ICNC$_{t0}$ exponent ($\beta_0 = 0.73$) is below 1, which is consistent with the above free power-law fitting, which includes
only one controlling factor, ICNC$_{t0}$. This strong, but nonlinear, positive contribution aligns with both the causal graph (Fig. 7) and SHAP analysis (Fig. 9), which also identified ICNC$_{t0}$ as the dominant predictor. The temperature coefficient ($\beta_1 = 0.18$) indicates a positive sensitivity to temperature, consistent with laboratory evidence (Hosler et al., 1957) as well as the causal graph and SHAP findings, though the effect is modest. Finally, the major size exponent ($\beta_3 = 0.16$) is relatively small, suggesting a weaker dependence on crystal size compared to ICNC$_{t0}$ or temperature. This may reflect either the limited variation
in crystal size across experiments, especially the colder cases (Fig. 2e) or the dominant influence of concentration effects under high-ICNC$_{t0}$ seeded conditions.

### 4.4   Sensitivity to temperature and ICNC$_{t0}$

We evaluated the sensitivity of predicted aggregation rates to temperature and ICNC$_{t0}$ by comparing predictions from CatBoost (CatB) and the physical equation against observations (Fig. 10). Three prescribed ICNC$_{t0}$ levels—$10^1$, $10^2$, and $10^3$
L$^{-1}$—were selected to represent conditions characteristic of stratus clouds (Gultepe et al., 2001), deep convection (Heymsfield and Willis, 2014), and secondary ice production in convective cloud systems (Korolev et al., 2020). To enable a meaningful comparison between predictions and observations, we stratified the measurements into three corresponding ICNC$_{t1}$ intervals: $(10^{0.5}, 10^{1.5})$, $(10^{1.5}, 10^{2.5})$, and $(10^{2.5}, 10^{3.5})$ L$^{-1}$. These intervals approximately align with the predicted ICNC$_{t0}$ levels and allow an assessment of the models' ability to capture the observed magnitudes and trends across representative regimes.

Both models reproduce the observed positive temperature dependence within the measurement domain ($-7.8$°C to $-4.7$°C; shaded region in Fig. 10) and exhibit physically plausible trends beyond. At all fixed ICNC$_{t0}$, predicted aggregation rates increased systematically with temperature, broadly consistent with the observations. Observed rates generally clustered around the predicted values at their respective ICNC levels, supporting the proposed scaling.

CatBoost predictions were smooth and closely aligned with observations at high and intermediate ICNC levels, but at low
ICNC they exhibited pronounced fluctuations at warmer temperatures (T $> -6$°C) and a nearly flat response at colder temperatures (T $< -6$°C; solid dark blue line in Fig. 10). This behaviour likely reflects the high variability of observations in this regime, as similar scatter is evident in the measured data, which can obscure systematic dependencies and limit the model's ability to learn consistent patterns. In contrast, predictions from the physical equation were stable and monotonic across the full ICNC and temperature ranges, reproducing the general trends well, though slightly overestimating aggregation rates at the
lowest ICNC and $T < -7$°C. The increased variability in low-ICNC observations suggests that other factors, such as habit composition, size distribution, or turbulence, may exert a proportionally stronger influence when ICNC is low.

Although evaluation based on RMSE and $R^2$ over the full test set indicated slightly better performance of CatBoost compared to the physical equation, Fig. 10 suggests that the two approaches perform similarly in most regimes, with the physical equation outperforming CatBoost in some cases (e.g., low ICNC at $T < -7$°C). This apparent discrepancy can be partly attributed to



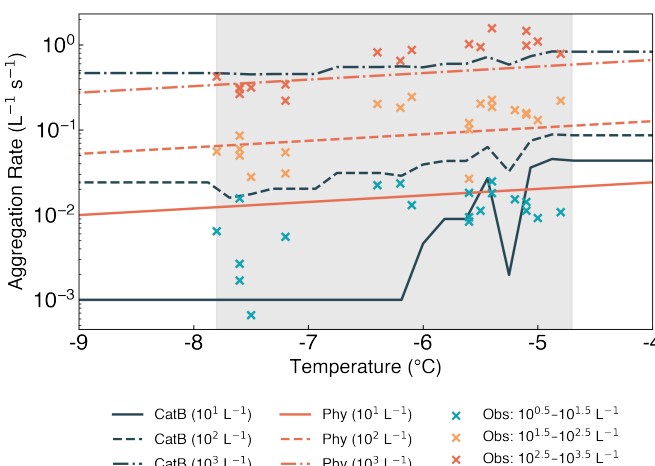

**Figure 10. Sensitivity of predicted and observed aggregation rates to temperature across ICNC levels.** Predictions from the CatBoost model (CatB) and a physical equation are shown for three prescribed initial ice crystal number concentrations ($ICNC_{t0}$) of $10^1$ (solid), $10^2$ (dashed), and $10^3$ L$^{-1}$ (dash-dotted); blue and red curves correspond to CatB and the physical equation, respectively. Observations are divided into three $ICNC_{t1}$ intervals: $(10^{0.5}, 10^{1.5})$, $(10^{1.5}, 10^{2.5})$, and $(10^{2.5}, 10^{3.5})$ L$^{-1}$, and are plotted as crosses in cyan, orange, and red, respectively. The shaded region indicates the observed temperature range ($-7.8°$C to $-4.7°$C).

how the performance metrics are computed: RMSE and $R^2$ are dominated by data-rich regions, where CatBoost tends to follow the observed scatter more closely—especially when the observations themselves are subject to variability from binning into three $ICNC_{t1}$ intervals $[(10^{0.5}, 10^{1.5})$, $(10^{1.5}, 10^{2.5})$, and $(10^{2.5}, 10^{3.5})$ L$^{-1}]$. In such regions, CatBoost can achieve lower residuals by adapting flexibly to local variations, whereas the physical equation prioritizes smooth, process-consistent behavior. In contrast, in sparsely sampled or more physically complex regimes, the physical equation yields more stable and

monotonic trends, while CatBoost shows larger deviations. These complementary behaviors underscore the value of combining data-driven flexibility with physically constrained formulations.

## 5 Conclusions

In this study, we present the in-situ quantification of ice aggregation rates in persistent supercooled stratiform clouds. This was made possible by a novel experimental design combining glaciogenic cloud seeding, which provided a controlled initial

state and allowed us to determine the age of ice crystals from nucleation to observation, and IceDetectNet, a deep-learning



algorithm capable of counting the number of monomers in individual ice crystals. This approach offered new insights into the microphysical and meteorological conditions that govern aggregation. The ICNCs observed during the experiments exceeded those typically found in naturally occurring stratiform clouds but were comparable to levels observed during secondary ice production events in convective clouds (Korolev et al., 2020). The combination of high ICNC and the absence of secondary

ice production provided a rare opportunity to isolate the contribution of aggregation without the confounding influence of ice multiplication processes and to generate observational constraints relevant to natural clouds.

We found $ICNC_{t0}$, temperature, major axis length, and aspect ratio as the primary controls on aggregation rate among all the factors we investigated, with $ICNC_{t0}$ emerging as the dominant factor. Even though $ICNC_{t0}$ has the strongest influence, a linear model based solely on $ICNC_{t0}$ failed to reproduce the observed rates, showing the importance of other factors and

the nonlinearity of their interactions. While $ICNC_{t0}$, temperature, and major axis length contribute positively to aggregation, aspect ratio acts as a negative factor, and riming showed no detectable effect, showing that riming and aggregation are largely independent processes. The influence of temperature operates both directly and indirectly by modifying ice crystal shape, consistent with current microphysical schemes (e.g. Seifert and Beheng (2006)). EDR showed no significant correlation with aggregation rate. This likely reflects our coarse resolution ($\sim$30 m) compared to the Kolmogorov and inertial subrange scales

($\sim$0.1 mm–10 m) where turbulence is expected to influence collisional growth (Pumir and Wilkinson, 2016), with additional masking by strong turbulence, narrow size distribution, and the dominant effects of $ICNC_{t0}$ and temperature.

We demonstrated that both machine learning models and the physically derived equation successfully reproduced the observed aggregation rates. CatBoost achieved the best performance in terms of RMSE and $R^2$ on the test dataset by capturing nonlinear interactions, whereas the physically based model proved more robust and stable in sensitivity tests, particularly in

regimes with low $ICNC_{t0}$, higher observational variability, and outside the observed temperature range. These differences highlight the complementary nature of data-driven and physically constrained approaches.

This study advances our understanding of the microphysical and environmental controls on ice aggregation. It emphasizes the central role of $ICNC_{t0}$ and the influence of temperature-dependent ice crystal properties. The study also showed that the aggregation occurs within 5-10 min after the ice formed. However, the mechanism behind the observed sub-quadratic relation-

ship between $ICNC_{t0}$ and aggregation rate remains uncertain. One possible explanation that is consistent with our observations is that aggregation may involve smaller ice crystals. However, this remains hypothetical. Addressing these uncertainties requires experimental designs that can capture the intermediate stages of ice crystal growth and interaction. The single-point experimental setup limits our ability to capture the full evolution of microphysical processes and may obscure intermediate stages of aggregation. Future experiments would benefit from a Lagrangian observational setup to better resolve the complete

aggregation pathway. This would allow us to refine the representation of aggregation in weather and climate models, ultimately improving predictions of precipitation and cloud radiative effects.

*Code and data availability.*  Data and scripts will be uploaded into a repository upon acceptance, and are available upon request until then.





**Table A1. Prediction uncertainty of IceDetectNet-CLOUDLAB.** Comparison between predicted (Pred) and hand-labeled (HL) monomer counts on the test set. Discrepancy (Discr.) is defined as prediction minus ground truth. Discr. (%) is normalized by HL Count.

| #HL Monomer | #HL Ice Crystal | Pred (mean $\pm$ std) | #Discr. | Discr. (%) |
|---|---|---|---|---|
| 1-monomer | 167 | $1.10 \pm 0.35$ | 16 | 9.6 |
| 2-monomer | 96 | $2.44 \pm 0.90$ | 21 | 21.9 |
| 3-monomer | 51 | $2.71 \pm 1.36$ | $-5$ | $-9.8$ |
| 4-monomer | 24 | $4.17 \pm 0.75$ | 1 | 4.2 |

## Appendix A: IceDetectNet-CLOUDLAB Uncertainties

To assess the uncertainty of IceDetectNet-CLOUDLAB, we compared its predicted monomer counts with manual annotations on a test set comprising 246 images of aggregates, totaling 359 labeled monomers. Discrepancies were calculated as the difference between predicted and annotated monomer counts. For example, if the hand label (HL) assigned two monomers while the prediction assigned one, the discrepancy was recorded as $-1$; if the prediction assigned three, the discrepancy was $+1$. Across all test images, the total discrepancy summed to 14, indicating a slight net overestimation of 14 monomers. To investigate the model's behavior across different aggregation levels, we grouped ice crystals by their hand-labelled monomer count (1 to 4) and computed the mean and standard deviation of the predicted counts within each group. Ideally, a perfect model would yield a mean equal to the true monomer count and a standard deviation of zero. The largest relative deviation occurred in the 2-monomer category, where the model overestimated the monomer count by 21.9% (TableA1). Predictions for 3-monomer aggregates showed a slight underestimation. For the most frequent class 1-monomer ice, ice crystals—IceDetectNet-CLOUDLAB performed reliably, with a mean prediction of 1.10 and a low standard deviation of 0.35 (TableF1). These results indicate that the model is robust for monomer number counting but tends to slightly overestimate complexity in multi-monomer aggregates.

## Appendix B: Correlation Between Aggregation Rate and EDR

We evaluated whether turbulence intensity, represented by EDR, influenced the aggregation rate across the 21 seeding experiments. No significant correlation was found in either the warmer ($r = 0.29$, $p = 0.321$) or colder ($r = -0.14$, $p = 0.767$) temperature regimes. Mean aggregation rates remained largely invariant across EDR bins, with substantial within-bin variability. HOLIMO imaged cloud particles within a three-dimensional volume of 11.76 cm$^3$ at 20 Hz during seeding conditions Fuchs et al. (2025); Ramelli et al. (2020, 2024), yielding an effective spatial resolution of approximately 1 m along the flight path. One likely explanation for the absence of a detectable signal is the sampling resolution: our measurements average over spatial scales far exceeding those at which turbulence is theoretically expected to influence collisional growth (Kolmogorov and inertial subrange scales of $\sim$0.1 mm–10 m) (Pumir and Wilkinson, 2016; Bodenschatz et al., 2010).




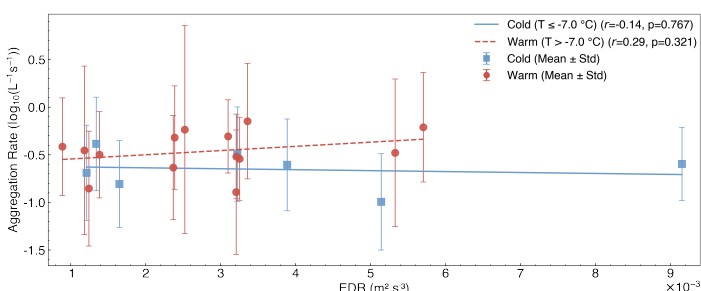

**Figure B1. Correlations between aggregation rate and EDR grouped by temperature.** Same as Fig. 2c, but with EDR as the x-axis.

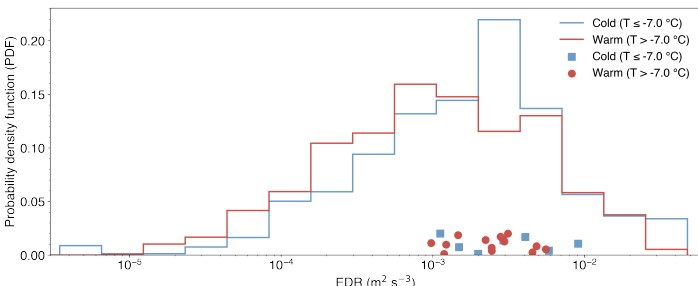

**Figure B2. Distribution of EDR grouped by temperature group.** Top: Probability density functions of EDR for all seeding experiments, grouped by temperature: colder ($T \leq -7°C$, blue) and warmer ($T > -7°C$, red). Bottom: Mean EDR values for individual experiments, shown as red circles (warmer) and blue squares (colder).

Nevertheless, spatial-scale limitations alone may not fully explain this discrepancy. Chellini and Kneifel (2024) reported enhanced aggregation and riming using Ka-band radar retrievals at comparable ~10–30 m scales, suggesting that resolution alone is unlikely to explain the absence of a turbulence signal in our data. Other factors may have contributed: (1) the seeded clouds were generally warmer, lower in altitude, and characterized by relatively strong turbulence. It is possible that EDR values were already sufficiently large for turbulence-driven aggregation effects to saturate, such that additional variability in EDR had little incremental impact; (2) the ice crystal size distribution in seeding experiments was relatively narrow and often dominated by specific ice habits, unlike the broader and more complex size distributions typical of natural mixed-phase clouds. This simpler distribution composition may reduce the sensitivity of aggregation processes to turbulence; and (3) the dominant influence of ICNC and temperature on aggregation rates could obscure any secondary turbulence effects, particularly in short-lived seeded clouds.





**Appendix C: Machine Learning Aggregation prediction Models**

We used a total of 10 supervised regression algorithms. Below we summarize the basic principles of each model:

– **Linear Regression** (Géron, 2022) fits a linear function by minimizing the squared difference between predicted and observed values. It assumes independent, additive relationships among input features and cannot capture nonlinear interactions.

– **Ridge Regression** (Géron, 2022) extends linear regression by adding $L_2$ regularization, which penalizes large coefficients and reduces overfitting, particularly in the presence of small sample sizes.

– **Bayesian Ridge Regression** (Géron, 2022) introduces a probabilistic prior over the regression coefficients and estimates them using Bayesian inference. This yields both regularized predictions and uncertainty estimates, while still assuming linearity.

– **K-Nearest Neighbors** (Géron, 2022) is a nonparametric method that predicts output values by averaging the labels of the $k$ training samples closest to the test point. Similarity is typically measured by Euclidean distance in the standardized feature space. Although simple and interpretable, the method becomes less effective in high-dimensional spaces.

– **Decision Tree Regression** (Géron, 2022) recursively partitions the input space by choosing feature-value thresholds that minimize the prediction error (typically mean squared error) at each split. The resulting model is a set of hierarchical "if–then" rules—for example, "if $ICNC_{t0} > 800\ L^{-1}$ and temperature $< -6°C$, then...". At each node, the best split is selected without looking ahead. While decision trees can capture nonlinear relationships and feature interactions, they tend to overfit if not regularized.

– **Adaptive Boosting** (Géron, 2022) constructs an ensemble of weak learners—models that perform only slightly better than random guessing—by sequentially reweighting the training samples. After each iteration, higher weights are assigned to mispredicted samples, forcing the next learner to focus more on difficult cases. The final output is a weighted sum of all learners. While this can substantially reduce bias, AdaBoost can be sensitive to noise and outliers.

– **Gradient Boosting** (Géron, 2022) also builds an ensemble of decision trees, but instead of adjusting sample weights, it fits each new tree to the residuals (i.e., errors) of the combined ensemble so far. This stage-wise additive approach allows the model to incrementally improve predictions and capture complex nonlinear dependencies.

– **Light Gradient Boosting Machine** (Ke et al., 2017) is an optimized gradient boosting implementation. It uses histogram-based feature binning, where continuous input features are discretized into fixed-width bins to accelerate training and reduce memory usage. Unlike traditional level-wise tree growth, LightGBM grows trees leaf-wise by expanding the leaf with the highest loss, leading to deeper, more specialized trees. These optimizations make LightGBM highly efficient on large datasets.



– **Extreme Gradient Boosting** (Géron, 2022) enhances standard gradient boosting with $L_1$ and $L_2$ regularization to penalize model complexity and prevent overfitting. It also supports parallelized training and handles missing values natively during tree construction. These improvements make it robust and scalable for structured data tasks.

– **Extremely Randomized Trees** (Géron, 2022) is an ensemble of decision trees where both the features and split thresholds are selected randomly at each node, rather than chosen based on an optimal impurity measure. This high degree of randomness reduces variance and helps avoid overfitting, especially when the data contain noise.




**Appendix D: Correlation Between Aggregation Rate and ICNC**

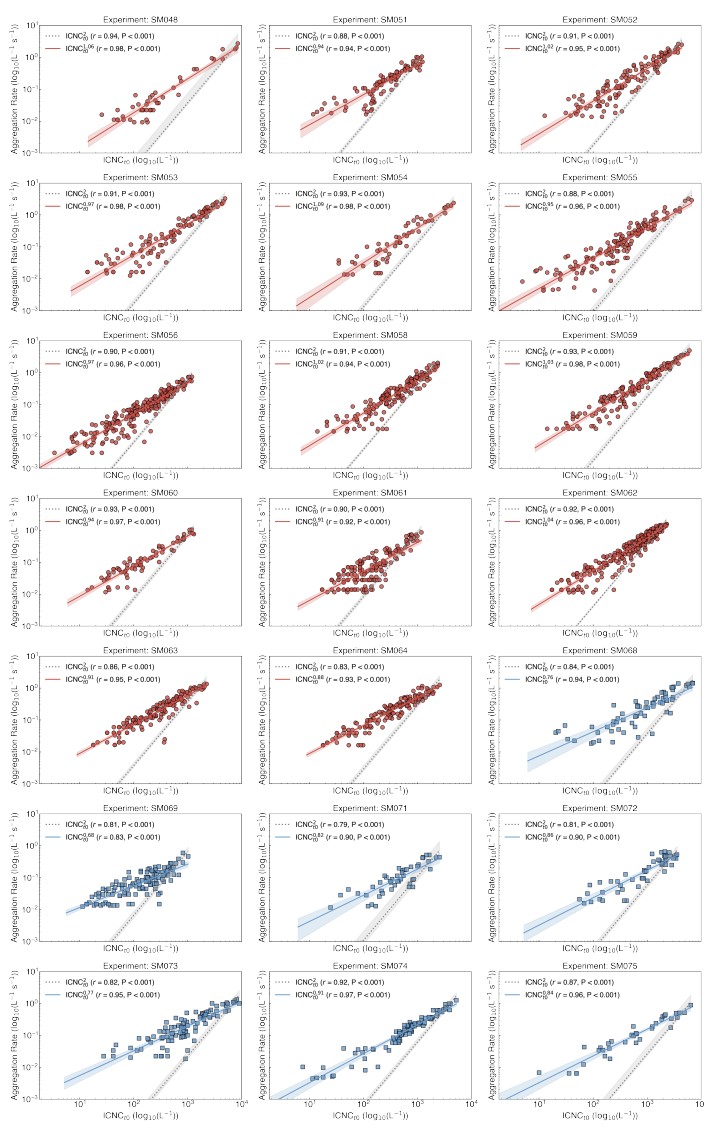

**Figure D1. ICNC$_{t0}$–aggregation rate relationships across all experiments.** Same as Fig. 3, but shown individually for all 21 experiments.

Red and blue markers represent warmer ($T > -7°C$) and colder ($T \le -7°C$) cases, respectively.



**Appendix E: Examples of ice crystals**

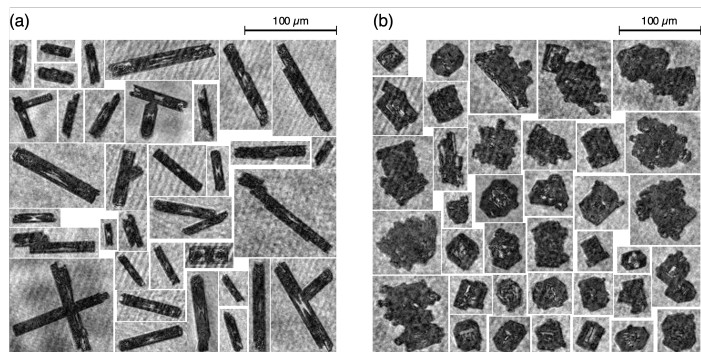

**Figure E1. A randomly selected sample of ice crystal images observed by HOLIMO during** (a) SM054 and (b) SM069



## Appendix F: Experiments overview

**Table F1. Summary of all seeding experiments.** For each experiment (Exp.) (where 'SM' denotes seeding mission, consistent with (Fuchs et al., 2025; Miller et al., 2025)), the table lists the number of 1-s data points(#Data), residence time between seeding and sampling (Time, in s), wind speed at seeding level (Wind, in m s$^{-1}$), temperature (Temp, in (°C)), mean ice crystal number concentration (ICNC, in cm$^{-3}$), background liquid water content (LWC, in mg m$^{-3}$), cloud droplet number concentration (CDNC, in cm$^{-3}$), and mean eddy dissipation rate (EDR, in $10^{-3}$ m$^2$ s$^{-3}$). ICNC, LWC, CDNC, and EDR reported values are means $\pm$ standard deviations.

| Exp. | #Data | Time (s) | Wind (m s$^{-1}$) | Temp (°C) | ICNC (L$^{-1}$) | LWC (mg m$^{-3}$) | CDNC (cm$^{-3}$) | EDR ($10^{-3}$ m$^2$ s$^{-3}$) |
|---|---|---|---|---|---|---|---|---|
| SM048 | 61 | 650 | 4.0 | -5.6 | 563 ± 1388 | 45 ± 34 | 106 ± 83 | 1.2 ± 0.9 |
| SM051 | 138 | 358 | 7.2 | -4.8 | 110 ± 127 | 107 ± 25 | 178 ± 43 | 4.5 ± 5.5 |
| SM052 | 174 | 430 | 6.0 | -5.6 | 308 ± 458 | 228 ± 34 | 369 ± 76 | 2.4 ± 2.3 |
| SM053 | 146 | 429 | 6.0 | -5.1 | 370 ± 458 | 226 ± 37 | 319 ± 77 | 5.6 ± 5.4 |
| SM054 | 64 | 464 | 4.5 | -5.0 | 370 ± 852 | 267 ± 27 | 295 ± 40 | 2.4 ± 1.9 |
| SM055 | 178 | 398 | 5.2 | -5.1 | 247 ± 501 | 235 ± 28 | 275 ± 32 | 4.8 ± 5.0 |
| SM056 | 261 | 560 | 5.5 | -5.2 | 75 ± 101 | 215 ± 24 | 267 ± 47 | 1.2 ± 1.2 |
| SM058 | 191 | 489 | 5.3 | -5.5 | 191 ± 206 | 229 ± 14 | 423 ± 35 | 1.0 ± 1.1 |
| SM059 | 175 | 403 | 5.1 | -5.4 | 325 ± 443 | 249 ± 18 | 471 ± 48 | 3.0 ± 4.6 |
| SM060 | 101 | 519 | 5.9 | -5.4 | 85 ± 124 | 261 ± 17 | 453 ± 74 | 2.2 ± 1.5 |
| SM061 | 350 | 619 | 4.1 | -5.6 | 54 ± 61 | 450 ± 37 | 437 ± 42 | 2.9 ± 5.3 |
| SM062 | 320 | 629 | 4.1 | -6.1 | 245 ± 203 | 383 ± 54 | 380 ± 60 | 2.8 ± 5.3 |
| SM063 | 184 | 541 | 3.8 | -6.4 | 180 ± 209 | 394 ± 43 | 407 ± 79 | 1.5 ± 2.2 |
| SM064 | 165 | 550 | 3.7 | -6.2 | 233 ± 251 | 348 ± 28 | 357 ± 63 | 3.1 ± 3.3 |
| SM068 | 99 | 339 | 7.6 | -7.8 | 851 ± 1387 | 239 ± 18 | 345 ± 38 | 2.0 ± 2.7 |
| SM069 | 177 | 474 | 6.5 | -7.6 | 98 ± 128 | 300 ± 15 | 345 ± 36 | 5.8 ± 9.1 |
| SM071 | 80 | 532 | 4.8 | -7.6 | 267 ± 397 | 277 ± 22 | 310 ± 44 | 1.5 ± 1.7 |
| SM072 | 94 | 331 | 7.7 | -7.5 | 599 ± 857 | 275 ± 38 | 352 ± 69 | 9.1 ± 13.2 |
| SM073 | 189 | 313 | 8.2 | -7.6 | 663 ± 1387 | 115 ± 27 | 121 ± 33 | 2.9 ± 2.5 |
| SM074 | 113 | 371 | 8.2 | -7.2 | 573 ± 726 | 148 ± 21 | 172 ± 28 | 4.1 ± 3.7 |
| SM075 | 45 | 294 | 7.0 | -7.2 | 1064 ± 1938 | 210 ± 28 | 191 ± 31 | 1.1 ± 1.4 |

*Author contributions.* H.Z. performed the scientific analysis, prepared the figures, and wrote the manuscript. H.Z. (SM048–SM062, except SM055 and SM056), C.F. (SM063–SM073), and F.R. (SM055, SM056, SM074, SM075) analyzed the data from the holographic imager. H.Z., C.F., F.R., A.J.M., N.O., R.S., and J.H. performed the seeding experiments and in situ measurements. H.Z. and Y.C. manually labeled the ice crystals used for training IceDetectNet-CLOUDLAB. H.Z. and X.L. developed the original IceDetectNet model and designed the



training strategy for IceDetectNet-CLOUDLAB. H.Z. trained 11 machine learning models, with some hyperparameter suggestions from

X.L. H.Z., J.H., and U.L. designed the physical model, and H.Z. performed the training. Z.W. calculated the eddy dissipation rate. U.L., J.H., and F.R. supervised the analysis and provided scientific guidance. U.L., J.H., and F.R. conceived CLOUDLAB and secured project funding. All authors contributed to revising the manuscript and approved the final version.

*Competing interests.* The authors declare that they have no conflict of interest.

*Acknowledgements.* We acknowledge financial support from the European Research Council (ERC) under the European Union's Horizon

2020 research and innovation program (grant no. 101021272). We would also like to extend our sincere gratitude to the following colleagues and collaborators: From the ETH domain, Benjamin Krummenacher for developing the deep learning algorithm to classify ice crystals as aggregated or not as part of his Bachelor thesis with H.Z.. From TROPOS, we thank Patric Seifert for providing the vertically pointing and scanning Doppler cloud radars and Kevin Ohneiser for maintaining the instrumentation. We are also grateful to Robert Oscar David (UiO) for many insightful scientific discussions.



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
