# Peer review of "Inferring the Controlling Factors of Ice Aggregation from Targeted Cloud Seeding Experiments"

_EGUsphere, 2025_

## Author Comment (AC1)

**Authors' comments to Anonymous Referee #1**

We would like to thank the reviewer for the thorough review of our manuscript and insightful feedback. These comments have significantly improved the quality of our work. In the following sections, we present the reviewer's comments (in black), our responses (in red), and the changes made in the revised manuscript (in blue). Please note that all line numbers in our responses correspond to those in the revised manuscript.
* * *
**Overall comments:**

I have thoroughly read the article titled "Inferring the controlling factors of ice aggregation from targeted cloud seeding experiments" by Zhang et al. Overall, I find the article very interesting and of high scientific quality although slightly limited due to experimental range. The article reports on the quantification of ice particle aggregation rate in cloud with several controlling factors in mind. They use machine learning to identify the main controlling factors. Although the study is very novel and the results are of high interest to the scientific community, there is an insurmountable modest dissatisfaction because the experimental space is so limited (the temperature range is limited and liquid droplets are necessarily present in high concentrations). These limitations only moderately take away from the novelty of the experimental design and the quality of the analysis. Overall, I feel that the article is definitely suitable for publication in ACP with some comments below. My main concern is about the utility of the results. While it is spectacular that the results have been so well studied, the authors should consider more closely where the results will be used and in what form it is best for those who will use them. Aggregation rates are used in high resolution models. The final results presented, while they may be the most accurate based on the data, produce a moderately rough line. I feel the article would be far more useful if the authors were to present suggestions as to how the results could be used in a model.

  Overall criteria:
  Scientific significance (4 Excellent): The publication tackles a very significant
      unknown through experiments. Aggregation rate of ice crystals in a cloud is a
      very difficult, yet important, number to know. This publication presents the
      results of the aggregation rate estimate after an extensive field experiment which
      attempts to accurately measure the range of possible values in the real

world.  The experimental design has been well documented. The analysis of the results (the heart of this publication) stand up in quality to the experimental design.  The reason I don't give it a 5 is that the significance would be higher if they had been able to calculate aggregation rate in a wider range of temperatures which was likely limited by experimental constraints.

Scientific quality (4 Excellent): Overall, the scientific quality of the research is excellent.  The authors use the latest in Machine Learning techniques to get at the analysis.  See specific comments regarding areas where the scientific quality could be improved.

Presentation quality (4 Excellent):  The manuscript is, for the most part, concise and the figures are well presented and easy to interpret.  English grammar is perfect. The authors have done a good job moving some details into Appendices which makes the article read well.

Specific comments:
1. Line 10-11:  In many microphysics parameterizations, an uncertainty of 0.08 might be considered "close enough" to 1 to be within the margin of error.  I would suggest presenting a level of confidence here to help the reader understand that you are very confident that it is not 1.0 and random experimental factors drove your mean to be below 1.
   Thanks for this suggestion. We now report the uncertainty of the ICNC exponent: across the 21 experiments, the mean exponent is 0.92 (standard error 0.024), yielding a 95% confidence interval of 0.876–0.968, which does not include 1.0. This indicates that the sub-quadratic scaling is statistically significant rather than an artefact of experimental variability.
   **L10-11:**
   We report, however, a subquadratic dependence of the aggregation rate on $ICNC_{t0}$ (mean exponent ~0.92; 95% CI: 0.88–0.97), in contrast to theoretical expectations (quadratic dependence).

2. Line 19: Don't forget that ice crystals do grow from vapor in regions that do not include supercooled liquid drops such as in cirrus at -50C where WBF can't exist.

We agree that vapor depositional growth also occurs in fully glaciated clouds where the WBF mechanism is not active, such as cirrus at very cold temperatures and have the revised the manuscript as follows:
**L18-20:**
During the early stage of ice growth, diffusional processes such as the Wegener–Bergeron–Findeisen mechanism dominate (Korolev, 2007) and vapor deposition in fully glaciated clouds such as cirrus (Gierens et al., 2003).

3. Line 100: Since EDR wasn't an important factor, perhaps it might be easier to reduce the text here and just say that EDR wasn't an important factor rather than giving the details on how EDR was measured.
Thank you for the suggestion. We have simplified the description of the EDR retrieval by retaining only the methodological essentials and removing detailed instrument-specific parameters.
**L103-105:**
Primary estimates were obtained from the Mira35 MBR7 Ka-band radar. In addition, the RPG94 W-band radar was used to supplement data gaps. The retrieval was temporally averaged over 30 s, and the resulting EDR fields have a spatial resolution of approximately 30 m.

4. Line 124: smaller crystals rather than smaller ones.

Fixed, thanks!

**L122-123:**
The uncertainty in cloud droplet number concentration is approximately ±5 %, while that for ice crystal number concentration ranges from 5–10 % for crystals larger than 100 µm and about 15 % for smaller ice crystals.

5. Line 146: Please use terms such as "variability" and "confidence" when possible rather than "uncertainty" (which is interpreted by the general public as "we don't really know").
We appreciate the reviewer's suggestion regarding the use of "uncertainty" in contexts where the term may be misinterpreted. In the specific paragraph noted, however, we are referring to potential detection errors of IceDetectNet rather than statistical variability or confidence levels. In this technical context, "uncertainties" denotes possible misidentification of monomers and is the

standard terminology for model- or algorithm-related error sources. To avoid confusion, we revised the text to use more precise terms.

**L149-150:**

Other potential detection errors and classification errors are discussed in Appendix A and are considered negligible.

6. Line 172: While you eliminate EDR as being too important, you might comment on measured EDR versus what would be expected in deep convection. (The same for other parameters, knowing how your data fit into the zoo of cloud types will help the reader to understand how representative your results are of situations of interest to the reader)

Thank you for the suggestion. We have revised the manuscript to place the observed EDR values primarily in the context of other boundary-layer cloud environments, which are dynamically more relevant to the clouds sampled in this study. For reference, we also note that EDR values reported for deep convective environments are typically one to two orders of magnitude higher (typically $10^{-2}$–$10^{-1}$ m² s⁻³; Barber et al. 2019), highlighting that the turbulence levels encountered here are comparatively weak.

**L188-191:**

EDR values measured during the seeding flights were modest (on the order of $10^{-4}$–$10^{-3}$ m² s⁻³, Fig. B1), comparable to turbulence levels reported for other boundary-layer cloud environments (Chu et al., 2025). This indicates that aggregation developed under a weak-turbulence conditions characteristic of stratiform mixed-phase clouds.

7. Figure 2: Are there symbols missing from the top? There is a blue line, then it gives the Cold temperature range twice, then a red line and warm twice. I suspect there should be a symbol there but it could be my computer.

Fixed, thank you!

8. Line 240 and paragraph below: Since there is a significant habit transition between warm and cold, it might be interesting to see if there are enough data points in each of the groups to identify a trend in each. There might be a natural functional change due to habit that could be hidden by linear analysis.

Thanks. In response, we quantified the number of observations in each temperature–habit group. The colder regime ($T \leq -7°C$) contains 797 data points

from 7 experiments, and the warmer regime ($T > -7°C$) contains 2508 data points from 14 experiments. Both groups therefore have sufficient sampling to examine trends within each habit regime. We now report these sample sizes explicitly in the revised manuscript.

**L169-170:**

Based on this habit difference, the experiments were classified into "warmer" (T > −7 ◦C; 14 experiments, 2508 data points) and "colder" (T ≤ −7 ◦C; 7 experiments, 797 data points) regimes (Fig. 2a).

9. Line 256: RR can influence geometry and stickiness if there is still some quasi liquid present?

Thanks. Indeed, riming can potentially modify not only ice crystal geometry but also the stickiness of colliding. We have added this clarification to the revised text.

**L268-270:**

Because riming influences ice crystal geometry — particularly major size and aspect ratio — and it may also influence surface stickiness, especially shortly after the riming event, it can, in principle, affect aggregation indirectly.

10. Line 275: I am not sure that I see how RR decreases ICNC_t0.

Thanks for pointing this out. We agree that riming cannot physically reduce $ICNC_{t0}$, and the RR → $ICNC_{t0}$ link in the preliminary DAG does not represent a causal mechanism. The weak negative coefficient arises from a statistical dependence: cases with higher ICNC exhibit lower riming efficiency per crystal because the available supercooled liquid water is shared among more particles. This resource-competition effect induces a negative correlation even though the causal direction is the opposite, and the link is not relevant to the aggregation-rate analysis. We therefore removed the RR → $ICNC_{t0}$ arrow from the final figure and clarified this point in the text.

**L290-295:**

RR has a negligible direct effect on aggregation (−0.01). RR indirectly increases the major axis length (+0.31), which tends to promote aggregation; however, the positive contribution of major size to the aggregation rate is itself very small (+0.08).

RR has a negligible direct effect on aggregation (−0.01). RR indirectly increases the major axis length (+0.31), which tends to promote aggregation; however, the positive contribution of major size to the aggregation rate (+0.08) is itself very small. Moreover, most rimed ice crystals in our measurements exhibit only light riming (Fig. E1), limiting the potential for riming-induced enhancements in fall speed or sticking efficiency. Under these specific experimental conditions, the overall influence of RR on aggregation appears minimal, suggesting that any coupling between riming and aggregation is weak or not detectable within these experiments.

11. Line 300: It would be nice to have a reference for the SHAP calculations.

Thanks. Reference added.

**L316-317:**

Model interpretability was assessed using SHapley Additive exPlanations (SHAP; Lundberg and Lee 2017).

12. Figure 10: The obvious question is what happened at -5.3 degrees? The CatB values all dip there especially the $10^1 L^{-1}$ line, the other CatB lines are impacted there as well. As stated in the initial comments, how is a modeler supposed to incorporate these data into a weather forecasting model?

Thanks. The feature near −5.3 °C reflects the higher intrinsic variability of aggregation rates at low ICNC rather than data sparsity. This has now been quantified and clarified in the revised manuscript using the coefficient of variation.

**L428-439:**

This behavior reflects the high variability of observations in the low-ICNC regime. We quantify this variability using the coefficient of variation (CV), defined as $CV = \sigma/\mu$ where $\sigma$ and $\mu$ are are the standard deviation and mean of Ragg within a given bin. Across the observed temperature range, CV values at low ICNC are typically 1.2--2.1, compared to 0.7--1.0 at intermediate ICNC and $\lesssim 0.6$ at high ICNC. Such large CV values indicate that aggregation rates in this regime are dominated by strong fluctuations rather than a clear temperature signal. In combination with the imposed lower bound on the target variable during training, this lack of a resolvable signal causes the tree-based model to revert to the lower-bound baseline, resulting in an approximately constant prediction at low temperatures. The flat behavior therefore reflects the absence of a learnable temperature dependence in this regime, rather than a physically meaningful relationship. The fact that a localized dip near -5.3°C appears consistently across

all ICNC regimes, while being most pronounced at low ICNC and progressively weaker at higher ICNC, further supports this interpretation. This feature is more likely to reflect experiment-to-experiment variability and condition mixing near this temperature, together with the non-smooth, threshold-based nature of tree models, rather than a robust, generalizable physical transition.

**L443-446:**

From a modeling perspective, this implies that physically based parameterizations provide a more reliable baseline representation of aggregation under such conditions, with observational or machine-learning-based approaches best suited for diagnosing variability or informing uncertainty estimates rather than defining deterministic temperature dependence.

13. Lines 491-526: I think it would be reasonable to add the Geron, 2022 reference on line 491 and not have it on every (but one) of the model algorithms.
Thank you for this suggestion. We agree that citing Géron (2022) repeatedly for individual algorithms is unnecessary. We now only cite Géron (2022) once at the beginning of the section.
**L522-523:**
We used a total of 10 supervised regression algorithms. A general introduction to these machine learning methods can be found in Géron (2022). Below we summarize the basic principles of each model:

**Reference:**

Barber K A, Deierling W, Mullendore G, et al. Properties of convectively induced turbulence over developing oceanic convection[J]. Monthly Weather Review, 2019, 147(9): 3429-3444.

Lundberg, S. M. and Lee, S.-I.: A unified approach to interpreting model predictions, Advances in neural information processing systems, 30, 2017.

---

## Author Comment (AC2)

**Authors' comments to Anonymous Referee #2**

We would like to thank the reviewer for the thorough review of our manuscript and insightful feedback. These comments have significantly improved the quality of our work. In the following sections, we present the reviewer's comments (in black), our responses (in red), and the changes made in the revised manuscript (in blue). Please note that all line numbers in our responses correspond to those in the revised manuscript.
* * *
**Overall comments:**

This is a really interesting paper, and it is wonderful to see these innovative experiments in a field which has been largely 'stuck' for many years. The authors use a novel experiment where they seed supercooled clouds to produce a high concentration of ice crystals, and then observe the outcome of the aggregation of those ice crystals a short time later. I do have quite a few comments which I outline below. I hope they are helpful. The main weakness for me at the moment is that some of the arguments about the interpretation of the data and the physical meaning of the derived aggregation rate are a bit 'fuzzy' and this leaves the reader feeling uncertain at times whether the conclusions drawn by the authors are really supported by the evidence. I think this could be tightened up and would make the paper more impactful. Using correlation between variables to draw conclusions and develop a physical model can be tricky, particularly if the underlying behaviour is obscured by random variability, and their are some places where the authors could perhaps rethink some of their conclusions on this (details below). I will state at this point that my knowledge of machine learning is very limited, and I have given less scrutiny to section 4.2 than to other parts of the manuscript.

Specific comments:
  Section 2.1
  1. On first reading I got a bit confused about where the sampling was happening following seeding. From figure 1 it looks like this occurs at essentially the same height as where the seeding occurred. Is this correct? If so, what about settling of the particles produced, and their aggregates? In a water saturated environment crystals grow pretty fast, and might fall of order 100m fall distance in time t. What effect might this have on the analysis?

2. You say "sedimentation losses within the plume are balanced by ice crystals falling from above, which is supported by radar observations" Can you show this supporting evidence?

Response to Q1 and Q2:

Thanks. Sampling occurs at approximately the seeding height. Although individual ice crystals do experience sedimentation, the plume passage time scale is short, such that sedimentation mainly results in vertical redistribution rather than a net removal of ice from the sampling volume. Radar observations do not indicate a systematic downward displacement of the enhanced reflectivity associated with the seeded plume, consistent with the absence of a strong sedimentation-driven loss. Under these conditions, treating the total number of ice crystal monomers as approximately conserved is a reasonable first-order approximation for the aggregation-rate analysis.

**L141-144:**

This implies that (i) sedimentation losses within the plume are approximately balanced by ice crystals falling from above. This assumption is supported by radar observations, which do not show a systematic downward displacement of the enhanced reflectivity associated with the seeded plume during the observation period (see Fig. 2 in Fuchs et al. (2025)).

Section 2.2

3. I liked the way that you have attempted to isolate aggregation from other growth processes in the cloud by assessing the number of monomers in the aggregates, as opposed to looking at the evolution of particle size over time, that seems like a clever innovation. Has this been done in any previous literature?

Ice-crystal aggregation and aggregation efficiency have been investigated in previous laboratory and modelling studies, for example using cloud chambers or particle size distribution (PSD) evolution to infer aggregation rates or sticking efficiencies (e.g. Connolly et al. (2012), as noted in Line 56 of the manuscript). Such studies typically rely on bulk metrics—such as changes in mass, number concentration, or PSD shape over time—combined with microphysical modelling. To the best of our knowledge, however, previous studies have not combined a controlled in-cloud seeding experiment with an image-based detection algorithim capable of resolving and counting individual monomers within observed aggregates. While modelling studies such as Karrer et al. (2020) examined the role of monomer number in determining aggregate properties, these analyses

were purely simulation-based and not applied to in situ observations with a known initial ice-crystal population. In contrast, our approach exploits the unique combination of a well-defined seeding event from CLOUDLAB and the IceDetectNet algorithm (Zhang et al., 2024), enabling monomer-level characterization of aggregates observed in real clouds. This provides a direct way to reconstruct the initial ice crystal number concentration and isolate aggregation from other growth processes over a known residence time.

**L64-65:**

…., providing a monomer-resolved level of detail for quantifying aggregation.

4. I do think you could explain equation 1 with greater clarity. It's the cornerstone of the analysis, so the reader should be confident in it, and it took me a while to make sense of. For example, you say that "Each detected aggregate containing ni monomers was assumed to have undergone (ni−1) aggregation events and have ni ice crystals at the initial state t0." Is this really an assumption? If we are saying that everything is a monomer at t = t0 and that you observe ni monomers in your aggregate at t = t1, then is there any way that it couldn't have been produced by (ni − 1) aggregation events? (as long as they are all binary collisions)

Thanks for this helpful clarification. We agree that describing the relation $(n_i-1)$ as an assumption was imprecise. We have revised the wording accordingly.

**L127-129:**

Under the definition that all ice crystals are monomers at the initial time $t_0$ and that aggregation proceeds via binary collisions, an observed aggregate containing $n_i$ monomers has undergone $n_i - 1$ aggregation events to reach its observed state at time $t_1$.

5. So then you define Ragg which "represents the mean aggregation rate integrated over the residence time". I think it's worth being slightly more precise - aggregation rate can mean a number of things depending on the context - I guess it is a mean number of aggregation events per second per unit volume over time interval t? (in fact, I noticed later that you write exactly this on line 203 later in the paper - can you bring this forward?)

We agree. We have clarified the definition of $R_{agg}$ at its first introduction and now explicitly state that it represents the mean number of aggregation events per unit time and per unit volume, averaged over the residence time $t$. This definition has been brought forward to improve clarity and to distinguish it from the instantaneous aggregation rate introduced later in the theoretical framework.

**L129-130:**

The aggregation rate Ragg (s−1 L−1) was thus defined as the mean number of aggregation events per unit time per unit cloud volume, averaged over the residence time t:

6. I found the terminology "residence time" a bit confusing. What do you mean by residence in this context? For example, in a lab experiment, you might think of residence time as a timescale beyond which all the particles fall out of the chamber. Here it just seems to be the time between seeding and measurement, so I'm not sure this is the right term to use. Maybe I misunderstood.

   Thank you for pointing this out. We switched the term from 'residence time' to 'advection time'.

   **L93-95:**

   The advection time t—the time between ice nucleation and observation—was calculated by comparing the times of seeding particles release and first ice detection, and using the observed wind speed across the plume to estimate advection time.

7. Is there an implicit assumption in the method here that Ragg is constant between t0 and t1? One thing you might worry about is that following nucleation there is a period of time when there is essentially no aggregation going on, because the crystals are tiny and have no relative fall speed. In this time they grow by diffusion. And then as they get bigger the aggregation rate ramps up, and we should expect from the structure of K (difference in fall speeds x mutual collision area) that the aggregation rate should increase as time goes on and the particles get bigger (either via aggregation itself, or via riming, deposition etc).

   We agree that, the aggregation rate is not expected to be constant between $t_0$ and $t_1$. Immediately after nucleation, ice crystals are small, have very limited differential fall speeds, and primarily grow by diffusion, so the aggregation rate is likely very low. As the ice crystals grow and the size distribution broadens, the

aggregation rate is expected to increase, as implied by the structure of the collision kernel.

We, however, do not attempt to resolve this time evolution. By construction, Eq. (1) defines $R_{agg}$ as a **time-averaged volumetric rate** over the interval between seeding and sampling. In other words, $R_{agg}$ represents the mean number of aggregation events per unit time and per unit volume over the finite time window sampled by the experiment, rather than the instantaneous aggregation rate at any particular moment. We have now clarified this definition in the revised manuscript.

**L129-131:**

The aggregation rate $R_{agg}$ ($s^{-1}$ $L^{-1}$) was thus defined as the mean number of aggregation events per unit time per unit cloud volume, averaged over the advection time t:

Section 2.4

8. In s2.4 you characterise the statistics of your seeded clouds, and estimate the ice crystal number concentration at time t0, which are in the ball park of several hundred crystals per litre. You then start to compare these numbers to natural (unseeded) clouds. I felt this was not very convincing. For example, when you say "These levels lie between typical ICNCt0 values reported for deep convection (Heymsfield and Willis, 2014) ($\sim 100 L^{-1}$) and those associated with secondary ice production in convective systems (Korolev et al., 2020) ($\sim 1000 L^{-1}$)", you seem to imply that in convective clouds without active secondary ice a typical ICNC is 100 per litre, which is not consistent with what I understand about. heterogeneous ice nucleation, nor with my reading of the Heymsfield and Willis paper, where they say in their discussion "The concentrations of this primary ice are on the order of $1 - 2 L^{-1}$". When there is ice multiplication, the number of ice particles can of course be much higher, and more similar to your experimental conditions. In general I think this comparison is not particularly compelling or useful: the purpose of seeding these clouds (presumably) is to control the number of ice particles by making them much more numerous than they would naturally be in that kind of cloud. I think the reader can probably accept that the aggregation process is the same, even if the concentration of particles is rather high, and I would favour making that argument a little more strongly, and toning

down the argument that what you have created is comparable to typical conditions in a natural system.

Thanks. We agree that the initial ice crystal number concentrations produced in our seeded clouds are not intended to represent typical primary ice concentrations in unseeded convective clouds, which are often much lower (of order $1-10\ L^{-1}$) in the absence of ice multiplication, as discussed by Heymsfield and Willis (2014). Our previous wording may have overstated the comparability between the seeded $\mathrm{ICNC}_{t0}$ values and those in natural clouds. We have therefore revised the text to remove the direct quantitative comparison with typical natural-cloud ICNC values and instead clarify that the aggregation process is governed by the same underlying collisional physics.

**L176-177:**

While these concentrations are higher than those typically found in unseeded clouds (Heymsfield and Willis, 2014), the aggregation process is governed by the same underlying collisional physics.

Section 3.1.1

9. On line 204 you make a connection between Ragg and the ICNC, by writing Ragg = 1 2 K(D1, D2, T)N1N2 and say "where N1 and N2 are the number concentrations of two ice crystal populations of sizes D1 and D2, T is temperature, and K(D1, D2, T) is the collision kernel, which depends on ice crystal size, shape, fall speed, and ambient conditions (Connolly et al., 2012)". It would like to encourage the authors to make this writing and arguments around this part of the study a bit more precise. What situation is being modelled here? The LHS is your mean rate of aggregation over some time interval t, averaged over all aggregates observed. The RHS is the instantaneous rate of aggregation of two types of particle (labelled 1,2) with each other. Are these things really equal? And under what assumptions? What do population 1 and 2 refer to in this context? And how does that relate to the formation of aggregates which may contain several monomers?

10. In the text you say K depends on the sizes and "ambient conditions" but in the equation we have the size and T. Is T the only relevant condition?

11. In what follows you say the simplest case is where the population is monodisperse, and then set N1 = N2 = ICNCt0 . An obvious objection to this is

that collisions are normally driven by difference in fall speeds, so if all the particles are the same, K = 0 and nothing happens. The other objection is that by definition the system cannot remain monodisperse while the crystals are aggregating; aggregation happens via collisions between clusters and monomers or one cluster + another cluster, and so then you need multiple terms on the RHS (with an awareness than Ni evolve over time)

12. I'm not trying to negate your overall argument about how Ragg should scale with ICNCt0 here: it's just that I don't think the text and equation convince the reader of the point you're trying to make. Interpretation of weak correlations.

**Response to comments 9-12:**
We agree that our original wording around Eq.~(2) was not sufficiently precise and could give the impression that the diagnosed aggregation rate is directly equivalent to the instantaneous kinetic aggregation rate.
In the revised manuscript, we now clearly distinguish between the **instantaneous aggregation rate** from kinetic theory and the **mean aggregation rate** $R_{\mathrm{agg}}$ derived from observations. The definition of mean aggregation rate is as clarified in response to comments 5 and 7, and the term *instantaneous* has been added for '**instantaneous aggregation rate'**.
We have also revised the description of the collision kernel by replacing "ambient conditions" with 'temperature'.
We agree that the earlier "monodisperse" formulation was misleading and physically inappropriate. In the revised text, this assumption has been removed and replaced by a more general scaling argument. Specifically, the kernel-based expression is now used only to motivate the expected scaling with $\mathrm{ICNC}_{t0}$.

**L209-218:**
… instantaneous aggregation rate (Seifert and Beheng (2006) Eq.62), …. K(D1, D2, T) is the collision kernel, which depends on ice crystal size, shape, fall speed, and temperature….In the simplest case, if the shape of the size distribution is approximately preserved and only its overall magnitude changes with the total ice number concentration, then the concentrations of the relevant size classes, N1 and N2, both scale linearly with the initial total ice concentration, ICNCt0. Under this approximation, both the instantaneous and time-averaged

aggregation rates are expected to scale as Ragg = K(D1, D2, T )(ICNC$_{t0}$)2. Motivated by this scaling,…

13. I encourage the authors to think about what is a meaningful correlation - i.e. something with a clear distinction from statistical independence / random noise. For example the connection between aggregation rate and temperature - I can draw a horizontal line through all of the data points (within their error bars) in Figure 5 - or even make a line with a negative slope. So I was not at all convinced that a 'real' correlation is present, even a weak one.

We agree that the evidence for a statistically robust temperature dependence is weak when assessed purely from correlation metrics. In the revised manuscript, we have therefore restructured and revised our words to avoid over-interpreting the correlation.

**L248-262:**

To investigate the role of temperature in shaping aggregation, motivated by the higher aggregation rate observed in the warmer case (SM054) compared to the colder case (SM069), we first examined how ice crystal size and geometry differ between warmer and colder experiments. Using the same representative cases shown in Fig. 3, we compared the warmer case SM054 (−5 ◦C) with the colder case SM069 (−7.6 ◦C). Consistent with known temperature-dependent ice habits, the warmer group (above −7 ◦C) consisted almost exclusively of columnar ice crystals (Fig. E1a), whereas the colder group (below −7 ◦C) included both columnar and plate-like crystals (Fig. E1b). Correspondingly, the warmer case exhibited consistently broader distributions of both major axis length and aspect ratio (Fig. 6). This behavior was not unique to these two cases: across the full dataset, warmer experiments showed broader distributions of major size (Fig. 2e) and aspect ratio (Fig. 2f) than colder experiments. Increased variability in crystal size and shape broadens the range of fall velocities among ice crystals (Heymsfield, 1972; Mitchell, 1996), which is expected to enhance collision frequency and thereby promote aggregation.

Consistent with this interpretation, aggregation rate shows a very weak positive association with temperature across all experiments (Pearson r = 0.43, p = 0.053; Fig. 5). While this relationship does not reach conventional significance levels and may therefore be difficult to distinguish from noise over the limited temperature range sampled, it is qualitatively consistent with the observed

structural differences in the ice population and with previous laboratory studies suggesting enhanced aggregation under warmer conditions (Hosler et al., 1957).

2.1 Section 3.2: causal graph

14. This causal graph process is described very briefly. This is not a standard method in cloud physics (to my knowledge), so is worth adding a little more detail to. You say "The graph structure was specified by combining prior physical knowledge with statistical dependencies inferred from the data". This implies there are some subjective choices required, so it would be good to say what these were and how you made those choices.

We agree that the description of the causal graph construction was too brief. In the revised manuscript, we add more descriptions and now clarify how the DAG structure was specified.

**L276-280:**

Specifically, candidate causal links were first proposed based on established microphysical understanding of ice aggregation (e.g., ICNC, size, and habit influencing collision and sticking probability), while arrows implying physically implausible causality (e.g., aggregation rate influencing temperature) were excluded a priori. Statistical associations were then used to assess whether the proposed links were supported by the data. The final DAG therefore reflects a physically constrained causal hypothesis rather than a data-driven structure learned automatically.

2.2 Dependence of aggregation rate on ICNC in models

15. In various places you allude to numerical model microphysics schemes and say that some of them assume aggregation rate scales linearly with ICNC. Could you make these links more explicitly please? These papers (Lin et al, Morrison and Milbrandt) are big, and also there are different forms of aggregation represented within them - for example in Lin et al aggregation between ice crystals is a very crude auto conversion arrangement, while snow - ice aggregation is an integration of the collection kernel over the PSD

We now clarify the precise links to operational microphysics schemes. In the Lin et al. (1983) scheme, the aggretaion rate of ice crystals is given by their Eq. (21):

$$P_{\text{SAUT}} = \alpha_1 (I_{CI} - I_0),$$

where $I_{CI}$ is the *ice mixing ratio* and $\alpha_1$ is a temperature-dependent rate coefficient and $I_0$ is a threshold. Thus, within this scheme, the aggregation tendency is **explicitly linear** in $I_{CI}$. To relate this to ICNC, we note that the scheme diagnostically links mixing ratio and number concentration via

$$q_i = N_i\, m_i,$$

where $m_i$ is the prescribed mass–diameter relation for ice particles. For a fixed mass–diameter parameterization (and the narrow size distributions typical of our seeded clouds), $m_i$ varies weakly, so $q_i$ scales approximately linearly with $N_i$. Therefore:

$$P_{\text{SAUT}} \propto I_{CI} \quad \Rightarrow \quad P_{\text{SAUT}} \propto N_i \quad \text{(i.e. ICNC)}.$$

This linear dependence contrasts with the quadratic dependence expected from collision-kernel theory for binary aggregation, and is the motivation for examining the empirical ICNC dependence in our data. We have added this clarification to the revised manuscript.

**L373-378:**
However, several bulk microphysics schemes adopt a linear dependence. For example, in Lin et al. (1983), the aggregation rate is parameterized as being proportional to the ice mixing ratio (Lin et al. (1983) Eq. 21). Because the mixing ratio is given by the product of number concentration and mean ice crystal mass, this formulation effectively yields a linear dependence on number concentration when the ice-crystal-mass–number relationship (i.e., the as
sumed PSD shape) is held fixed within the scheme. Similar linear forms also appear in two-moment parameterizations such as Morrison and Milbrandt (2015).

2.3 Parameterisation of temperature effect

16. I understand the idea of setting the temperature dependence in eq 2 as exp(β1T) but this strong nonlinearity seems hard to justify from your data, perhaps because your range of T is fairly limited. So I think this is worth thinking and discussing carefully.

17. You say "The temperature coefficient (ϵ1 = 0.18) indicates a positive sensitivity to temperature, consistent with laboratory evidence (Hosler et al., 1957) as well

as the causal graph and SHAP findings, though the effect is modest". It's modest in the small range of T you sampled. But if you applied exp(0.18T) to colder clouds (e.g. -15°C), the aggregation rate will be scaled down very sharply - and I think you need to caution the reader against extrapolation without further data to support that

Thanks. We agree that the exponential temperature term in Eq. (2) should be interpreted strictly within the limited temperature range sampled in our experiments ($-7.8$ to $-4.7°C$). Over such a narrow interval, different functional forms (e.g., linear, exponential, or weak power-law) are statistically difficult to distinguish, and the exponential term is used here as a flexible local representation rather than a globally valid physical law. However, it should not be extrapolated to substantially colder conditions, as this would imply an unrealistically strong temperature sensitivity. We have revised the manuscript to clarify that the fitted coefficient reflects behavior only within the observed temperature range and does not imply validity outside of it.
**L409-410:**
However, this coefficient reflects only the behavior within our observational temperature range (between $-4.7 \circ C$ and $-7.8 \circ C$).

2.4 The influence of riming

18. I was surprised that riming had no influence on the aggregation rate in your analysis. Riming enhances particle fall speeds (and their dispersion), which you would expect to drive collisions. You don't analyse riming initially in s3, but then your bring it in for the DAG. At this point you then dismiss it, since its (indirect) influences largely cancel out. However, I wondered: is this a consequence of the design of the graph? You don't allow RR to directly affect Ragg as far as I can see. Would the conclusion be different if you allowed that possibility? Or maybe there is not enough variation in RR in your dataset to evaluate this?

[Figure]

*Figure 1 Causal graph of factors influencing aggregation rate. The directed acyclic graph illustrates the inferred direct and indirect influences on the aggregation rate (R_agg) from five predictors: initial ice crystal number concentration (ICNC$_{t0}$), temperature (T), major axis length (MajSiz), aspect ratio (AR), and riming ratio (RR). Nodes represent variables, and arrows indicate causal pathways. Arrow thickness and labels denote the magnitude and sign of the standardized effects (change in R_agg in standard deviations per one-standard-deviation change in the predictor).*

19. In your conclusion you make a strong statement that "riming showed no detectable effect, showing that riming and aggregation are largely independent processes" I don't think you have the evidence to draw such a strong conclusion

   **Response to comments 18-19:**

   We agree that our earlier wording did not sufficiently clarify the riming conditions of our experiments. Most rimed particles in our dataset exhibit only light riming (Fig.12 in main text), which limits the extent to which riming can modify fall speeds or collision efficiencies. Under such conditions, any riming–aggregation coupling is expected to be subtle.

   Allowing RR to directly influence $R_{agg}$ in the DAG does not change this conclusion: the fitted direct coefficient remains very close to zero (–0.01, as shown in Fig 1), indicating that the data provide little support for a detectable

effect within our experimental range. We have now clarified this in the manuscript.

**L290-295:**

RR has a negligible direct effect on aggregation (−0.01). RR indirectly increases the major axis length (+0.31), which tends to promote aggregation; however, the positive contribution of major size to the aggregation rate (+0.08) is itself very small. Moreover, most rimed ice crystals in our measurements exhibit only light riming (Fig.12 in main text), limiting the potential for riming-induced enhancements in fall speed or sticking efficiency. Under these specific experimental conditions, the overall influence of RR on aggregation appears minimal, suggesting that any coupling between riming and aggregation is weak or not detectable within these experiments.

2.5 The influence of turbulence

20. You quickly determine that eddy dissipation rate from the radar is not correlated with aggregation rate and dismiss this as a parameter of interest for the rest of the paper, returning to it in the conclusions "EDR showed no significant correlation with aggregation rate. This likely reflects our coarse resolution ($\sim$ 30 m) compared to the Kolmogorov and inertial subrange scales. ($\sim$ 0.1 mm – 10 m) where turbulence is expected to influence collisional growth (Pumir and Wilkinson, 2016), with additional masking by strong turbulence, narrow size distribution, and the dominant effects of ICNCt0 and temperature.". In appendix B you make a similar statement: "our measurements average over spatial scales far exceeding those at which turbulence is theoretically expected to influence collisional growth". I don't really understand your argument here. It seems to imply the dissipation rate is scale-dependent - which of course it isn't (at least, within the usual Kolmogorov cascade framework). As long at your 30m radar box is within the inertial sub range (which I would suggest is likely), then there is not a problem.

21. Why are the Kolmogorov and inertial subrange scales the length scales where turbulence is expected to influence collisional growth? I would have expected the relevant scales to be related to the particles and their microphysical characteristics.

22. I also didn't understand your references to masking by "strong turbulence" and a narrow size distribution - please could you elaborate and make this more concrete, or remove if speculative.

23. You seem to not include the simpler possibility that aggregation is genuinely insensitive to dissipation rate, over the range of EDR that you sampled?

**Response to comments 20-23:**
We agree that our earlier explanation overly emphasized scale considerations and did not adequately clarify the physical reasoning. In the revised manuscript, we no longer attribute the lack of correlation to sampling resolution. Instead, we now discuss three physically grounded factors that may limit detectability of turbulence effects in our dataset: (i) the seeded clouds exhibit relatively strong background turbulence, such that aggregation-relevant turbulence effects may already be near saturation; (ii) the size distributions are narrow, reducing turbulence-driven differential fall speeds; and (iii) the dominant influences of $ICNC_{t0}$ and temperature can mask any secondary dependence on EDR in short-lived plumes.
We have also clarified that the simplest interpretation—that aggregation is only weakly sensitive to EDR over the dissipation-rate range sampled—cannot be ruled out. The revised text now reflects this more balanced interpretation.
**L474-476:**
EDR showed no significant correlation with the aggregation rate. This finding likely reflects the combined effects of narrow size distributions and the dominant roles of $ICNC_{t0}$ and temperature in these seeded clouds. Within the EDR range sampled, aggregation appears to be at most weakly sensitive to EDR, and any residual dependence is small compared with other controlling factors.

**L507-520:**
We evaluated whether turbulence intensity, represented by EDR, influenced the aggregation rate across the 21 seeding experiments. No significant correlation was found in either the warmer ($r = 0.29$, $p = 0.321$) or colder ($r = -0.14$, $p = 0.767$) temperature regimes. Mean aggregation rates remained largely invariant across EDR bins, with substantial within-bin variability. HOLIMO imaged cloud particles within a three-dimensional volume of 11.76 $cm^3$ at 20 Hz during seeding

conditions (Fuchs et al., 2025; Ramelli et al., 2020, 2024), yielding an effective spatial resolution of approximately 1 m along the flight path.

Several factors may contribute to the absence of a detectable EDR signal: (1) the ice crystal size distribution in seeding experiments was relatively narrow and often dominated by specific ice habits, unlike the broader and more complex size distributions typical of natural mixed-phase clouds, which may limit the ability to robustly identify turbulence-related effects in the available observations; and (2) the dominant influence of ICNC and temperature on aggregation rates could obscure turbulence effects, particularly in short-lived seeded clouds. A more straightforward interpretation is that aggregation is only weakly sensitive to EDR within the range of dissipation rates sampled here, and any remaining dependence is small relative to the dominant effects of $ICNC_{t0}$ and temperature.

**Reference:**

Connolly P J, Emersic C, Field P R. A laboratory investigation into the aggregation efficiency of small ice crystals[J]. Atmospheric Chemistry and Physics, 2012, 12(4): 2055-2076.

Karrer M, Seifert A, Siewert C, et al. Ice particle properties inferred from aggregation modelling[J]. Journal of Advances in Modeling Earth Systems, 2020, 12(8): e2020MS002066.

Lin Y L, Farley R D, Orville H D. Bulk parameterization of the snow field in a cloud model[J]. Journal of Applied Meteorology and climatology, 1983, 22(6): 1065-1092.

Morrison H, Milbrandt J A. Parameterization of cloud microphysics based on the prediction of bulk ice particle properties. Part I: Scheme description and idealized tests[J]. Journal of the Atmospheric Sciences, 2015, 72(1): 287-311.

Zhang H, Li X, Ramelli F, et al. IceDetectNet: a rotated object detection algorithm for classifying components of aggregated ice crystals with a multi-label classification scheme[J]. Atmospheric Measurement Techniques, 2024, 17(24): 7109-7128.